# 📐 Train Small, Infer Large: Memory-Efficient LoRA Training for Large Language Models

**Jun Zhang**[1,3]*, **Jue Wang**[1,3], **Huan Li**[1,2†], **Lidan Shou**[1,2†], **Ke Chen**[1,2],
**Yang You**[4], **Guiming Xie**[5], **Xuejian Gong**[5], and **Kunlong Zhou**[5]

[1]The State Key Laboratory of Blockchain and Data Security, Zhejiang University
[2]Hangzhou High-Tech Zone (Binjiang) Institute of Blockchain and Data Security
[3]College of Computer Science and Technology, Zhejiang University
[4]Department of Computer Science, National University of Singapore
[5]AI Center, Guangdong OPPO Mobile Telecommunications Corp., Ltd.
`{zj.cs,zjuwangjue,lihuan.cs,should,chenk}@zju.edu.cn`,
`youy@comp.nus.edu.sg`, `{xieguiming,gongxuejian,zhoukunlong}@oppo.com`

## Abstract

Large Language Models (LLMs) have significantly advanced natural language processing with exceptional task generalization capabilities. Low-Rank Adaption (LoRA) offers a cost-effective fine-tuning solution, freezing the original model parameters and training only lightweight, low-rank adapter matrices. However, the memory footprint of LoRA is largely dominated by the original model parameters. To mitigate this, we propose LoRAM, a memory-efficient LoRA training scheme founded on the intuition that many neurons in over-parameterized LLMs have low training utility but are essential for inference. LoRAM presents a unique twist: it trains on a pruned (small) model to obtain pruned low-rank matrices, which are then recovered and utilized with the original (large) model for inference. Additionally, minimal-cost continual pre-training, performed by the model publishers in advance, aligns the knowledge discrepancy between pruned and original models. Our extensive experiments demonstrate the efficacy of LoRAM across various pruning strategies and downstream tasks. For a model with 70 billion parameters, LoRAM enables training on a GPU with only 20G HBM, replacing an A100-80G GPU for LoRA training and 15 GPUs for full fine-tuning. Specifically, QLoRAM implemented by structured pruning combined with 4-bit quantization, for LLaMA-3.1-70B (LLaMA-2-70B), reduces the parameter storage cost that dominates the memory usage in low-rank matrix training by 15.81× (16.95×), while achieving dominant performance gains over both the original LLaMA-3.1-70B (LLaMA-2-70B) and LoRA-trained LLaMA-3.1-8B (LLaMA-2-13B). Code is available at **`https://github.com/junzhang-zj/LoRAM`**.

## 1 Introduction

Large language models (LLMs), such as GPT-4 (OpenAI, 2023), LLaMA (Touvron et al., 2023a;b; Dubey et al., 2024), and PaLM (Chowdhery et al., 2023), have recently revolutionized natural language applications. These models excel in task generalization, driven by their exponential increase in scale, with some exceeding 400 billion parameters (Dubey et al., 2024). Fine-tuning pre-trained LLMs is critical for task-specific customization, enhancing desired behaviors while mitigating undesired ones (Qi et al., 2024). However, this process is constrained by substantial memory requirements; for instance, fine-tuning a 70B LLaMA in 16-bit precision demands over 1178GB[1] of memory, necessitating an expensive setup of 15 GPUs (A100 or H100, each with 80GB HBM).

---

* Work done during an internship at OPPO AI Center.
† Huan Li and Lidan Shou are the corresponding authors.
[1]Training is performed on one sample with a length of 4K using `BF16` mixed precision with the Adam optimizer, incorporating gradient checkpointing.

To mitigate the high cost of fine-tuning LLMs, parameter-efficient fine-tuning (Li & Liang, 2021; Lester et al., 2021; Liu et al., 2021; Qiu et al., 2023; Liu et al., 2024b; 2022), particularly Low-Rank Adaption (LoRA) (Hu et al., 2022) and its variants (Liu et al., 2024a; Ding et al., 2023; Zi et al., 2023; Zhang et al., 2023b; Kalajdzievski, 2023), freezes the original LLM weights and only updates the injected low-rank matrices to adapt to new tasks under limited resources. However, during training, they still struggle with the significant memory footprint of the parameters of the base LLM, even with quantization (Dettmers et al., 2023; Xu et al., 2024; Li et al., 2024; Guo et al., 2024; Frantar et al., 2023; Chai et al., 2023). Typically, they reduce the precision to 4 bits at most due to quality considerations. This memory dilemma raises an interesting and necessary question:

*Can we further reduce the memory overhead of the base model during LoRA training while still maintaining the inference accuracy?*

Our answer is a resounding *Yes*! In this paper, we propose **M**emory-efficient **LoRA** training, coined LoRAM, a novel scheme to reduce the memory overhead of LoRA fine-tuning for LLMs. We revisit the training and inference process of the LoRA paradigm, building on it with a unique twist: Unlike typical LoRA, which uses the same original model for training and inference, LoRAM employs different models at each stage, i.e., it trains a *pruned (small)* model by updating the pruned LoRA weights and then performs inference on the *original (large)* model with the recovered low-rank matrices. This recovery process reshapes the pruned matrices to ensure be merged into the original model, allowing for the updating of unpruned weights while utilizing the pruned weights during inference. The key insight driving our approach comes from reconciling two seemingly contradictory concepts. The scaling laws (Hoffmann et al., 2022; Kaplan et al., 2020; Hernandez et al., 2021) suggest that a large number of parameters of LLMs is essential for effective model generalization.

Conversely, sparsity in LLMs (Zhang et al., 2024b; Ma et al., 2023; Sun et al., 2024; Frantar & Alistarh, 2023a; Xia et al., 2024) show that these pre-trained models can be compressed by removing redundant weights. The goal is to minimize the difference in the model's output before and after pruning. However, such methods tend to falter at higher pruning ratios and aggressively pruned models lose critical reasoning capabilities, e.g., only pruning 10%~20% (Ma et al., 2023; Frantar & Alistarh, 2023a). Our intuition builds on this: some critical parameters contribute significantly to fine-tuning, while other parameters, though essential for inference, usually remain unchanged during fine-tuning. Therefore, LoRAM leverages this insight by updating the weights retained through pruning from LoRA training (yellow blocks in Fig. 1) to significantly reduce memory usage and training time, while employing the pruned weights (blue blocks in Fig. 1) during inference to enhance generation performance (see Section 3.5).

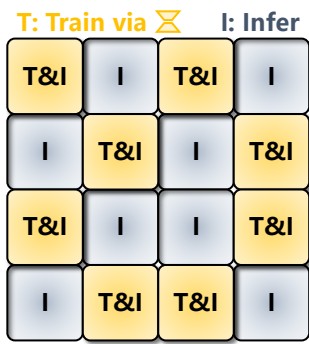

Figure 1: Idea of LoRAM

Despite the significant reduction in memory cost achieved by the pruning-recovery process of Lo-RAM, maintaining gains at more aggressive pruning rates (e.g., 65% or higher) remains challenging. We attribute this to the knowledge inconsistency between the pruned model used for training and the original model used for inference. To address this, we propose an effective alignment strategy: low-cost continual pre-training of the pruned model on a small dataset. This alignment is performed once offline, allowing the model's publisher to execute it. For instance, Meta AI could release a set of aligned pruned models for LLaMA-3, enabling low-resource users to fine-tune large models for customized tasks using LoRAM. Notably, as a bonus, LoRAM seamlessly integrates with existing quantization schemes designed for LoRA, such as QLoRA, forming QLoRAM, which further reduces memory overhead.

The contributions of this work are summarized as follows:

(1) *Novel Training Scheme*: We propose LoRAM, a memory-efficient LoRA training scheme. LoRAM trains a pruned model by updating the pruned low-rank matrices and then uses dimensionally recovered low-rank matrices to integrate with the original model for inference. The process significantly reduces the memory consumption incurred by the model parameters during training, and synergistically boosts performance by leveraging the full original parameters during inference. Thus, LoRAM efficiently enhances performance under limited device memory resources.

(2) *Effective Alignment Strategy*: We identify that the knowledge inconsistency between the pruned model used for training and the original model used for inference limits the performance gain of LORAM under aggressive pruning rates. To tackle this, we train the pruned model on a small amount of general corpus to achieve alignment, which is a one-shot offline process and can be easily performed by the model publisher.

(3) *Extensive Experimental Evaluation*: We conduct comprehensive experiments to validate the effectiveness of LORAM across various pruning algorithms, models of different sizes, and tasks in different domains. Notably, QLORAM which combines LORAM with structured pruning and 4-bit quantization reduces the memory cost of LLaMA-2-70B parameters by $16.95\times$ while effectively achieving performance gains superior to both the original LLaMA-2-70B and LLaMA-2-13B fine-tuned with LoRA.

## 2 MEMORY-EFFICIENT LoRA TRAINING — LORAM

### 2.1 LOW-RANK ADAPTATION

Given a pre-trained weight matrix $\mathbf{W}_0 \in \mathbb{R}^{m \times n}$, a typical full-parameter fine-tuning (Li et al., 2023b) process adapts to new tasks by updating the entire full-rank matrix $\mathbf{W}_0$. Inspired by the insight that pre-trained weights of LLMs exhibit low "intrinsic dimension" when adapting to specific tasks (Aghajanyan et al., 2021), LoRA (Hu et al., 2022) further suggests that the updated weights have a low "intrinsic rank". Consequently, LoRA reparameterizes the model weights as $\mathbf{W}_0 + \mathbf{W}_\Delta = \mathbf{W}_0 + \mathbf{BA}$, where $\mathbf{B} \in \mathbb{R}^{m \times r}$ and $\mathbf{A} \in \mathbb{R}^{r \times n}$, and $\mathbf{W}_\Delta = \mathbf{BA}$ represents a low-rank decomposition matrix with the rank $r \ll \min(m, n)$.

During training, as illustrated in Fig. 2 (a), the pre-trained weight matrix $\mathbf{W}_0$ is frozen to avoid gradient computation. Instead, the low-rank matrices $\mathbf{B}$ and $\mathbf{A}$ are updated to enable parameter-efficient fine-tuning, which defaults to standard supervised fine-tuning, with the objective function $\mathcal{L}_{\text{SFT}}$ defined as the cross-entropy loss between the predicted logits and the ground-truth answers. Given an input feature vector $\mathbf{x}$ of length $m$, the forward pass of LoRA modifies the output activation from fully fine-tuning, represented by $\mathbf{h} = \mathbf{x}\mathbf{W}_0$ (of length $n$), to:

$$\mathbf{h} = \mathbf{x}\mathbf{W}_0 + \mathbf{x}\mathbf{W}_\Delta = \mathbf{x}\mathbf{W}_0 + \mathbf{x}\mathbf{BA}. \tag{1}$$

Once low-rank matrices $\mathbf{B}^\star$ and $\mathbf{A}^\star$ are trained by minimizing the $\mathcal{L}_{\text{SFT}}$, as shown in the Fig. 2 (c), the computation of activation $\mathbf{h}$ for $\mathbf{x}$ is reformulated to improve inference efficiency:

$$\mathbf{h} = \mathbf{x}(\mathbf{W}_0 + \mathbf{W}_\Delta^\star) = \mathbf{x}(\mathbf{W}_0 + \mathbf{B}^\star \mathbf{A}^\star). \tag{2}$$

### 2.2 MEMORY-EFFICIENT LoRA TRAINING

Consider the LLaMA-2-13B model, we introduce low-rank matrices ($r = 8$) for the four projection matrices ($\mathbf{W}_q, \mathbf{W}_k, \mathbf{W}_v$, and $\mathbf{W}_o$) in the attention layer, the three projection matrices ($\mathbf{W}_{\text{up}}, \mathbf{W}_{\text{gate}}$, and $\mathbf{W}_{\text{down}}$) in the MLP layer, and the weight matrix $\mathbf{W}_{\text{lm\_head}}$ in output layer. Despite the additional 32 million parameters, the number of trainable parameters is reduced by $406\times$ compared to the full parameters. Many LoRA variats (Zhou et al., 2024; Zhang et al., 2023a; Kopiczko et al., 2024; Azizi et al., 2024; Wang et al., 2024) aim to address the significant memory overhead associated with $\mathbf{W}_\Delta$ as $\mathbf{W}_0$ scales, but they still necessitate storing a complete copy of $\mathbf{W}_0$ in memory, which dominates training memory usage. Even with quantization methods designed for LoRA (Dettmers et al., 2023; Xu et al., 2024; Li et al., 2024; Guo et al., 2024; Frantar et al., 2023; Chai et al., 2023), training performance constraints often limit the representation of $\mathbf{W}_0$ to 4 bits. Consequently, for the LLaMA-2-13B, storage requirements are reduced from 26 GB in `FP16` to 6.5GB in `NF4`. However, this is still significantly higher than the memory required for $\mathbf{W}_\Delta$ in `BF16`, which occupies only 64MB of storage and has a peak memory requirement of 576MB during training. Thus, the memory needed for the frozen quantized $\mathbf{W}_0$ is $11.5\times$ greater than that required for learnable $\mathbf{W}_\Delta$.

To mitigate the memory overhead dominated by $\mathbf{W}_0$ while achieving inference performance gain, we propose a memory-efficient LoRA training called LORAM. LORAM first prunes the model to a smaller size and performs LoRA fine-tuning on the pruned model. After training, it recovers the LoRA weights, applies them to the original model, and then conducts inference. We now describe the various stages of LORAM. The complete algorithm of LORAM is presented in Appendix F.

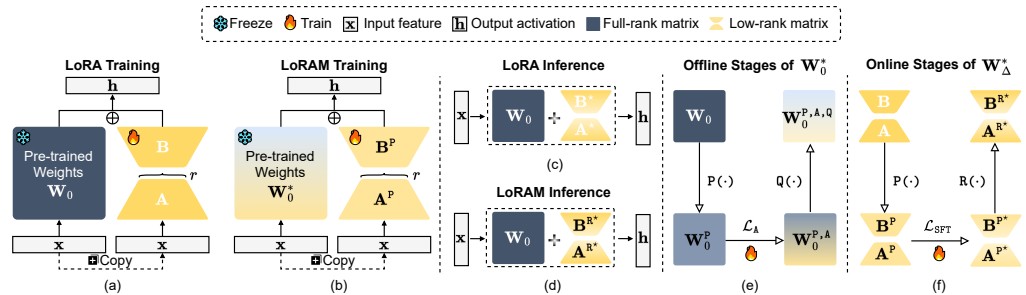

Figure 2: Comparison of LoRAM and LoRA: Training (subfigures a and b) and Inference (c and d). Key stages include the offline process of the frozen full-rank matrix $\mathbf{W}_0^*$ (subfigure e) and the online generation of the learnable low-rank matrix $\mathbf{W}_\Delta^*$ (f) during LoRAM training (b) and inference (d).

**Pruned Full-Rank Weight Generation.** First, we employ a pruning algorithm $\mathtt{P}(\cdot)$ to derive the pruned weight matrix $\mathbf{W}_0^\mathtt{P}$ from the original weights $\mathbf{W}_0$. Specifically, $\mathbf{W}_0^\mathtt{P}$ is computed as:

$$\mathbf{W}_0^\mathtt{P} = \mathtt{P}(\mathbf{W}_0) = \mathbf{W}_0 \circ \mathbf{M}^\mathtt{P}, \tag{3}$$

where $\mathbf{M}^\mathtt{P} \in \{0, 1\}^{m \times n}$ is a binary mask matrix indicating retained parameters ('1') and pruned parameters ('0'), and $\circ$ denotes the Hadamard product.

**Pruned Low-Rank Matrix Training.** After obtaining the pruned weight matrix $\mathbf{W}_0^\mathtt{P}$, we modify the standard LoRA training process. Instead of updating the low-rank matrices $\mathbf{B}$ and $\mathbf{A}$ for the original $\mathbf{W}_0$, we train the pruned low-rank decomposition matrix $\mathbf{W}_\Delta^\mathtt{P} = \mathbf{W}_\Delta \circ \mathbf{M}^\mathtt{P} = \mathbf{B}^\mathtt{P}\mathbf{A}^\mathtt{P}$, while keeping $\mathbf{W}_0^\mathtt{P}$ frozen as shown in Fig. 2 (b). The output activation $\mathbf{h}$ for an input feature vector $\mathbf{x}$ is calculated as:

$$\mathbf{h} = \mathbf{x}\mathbf{W}_0^\mathtt{P} + \mathbf{x}\mathbf{W}_\Delta^\mathtt{P} = \mathbf{x}\mathbf{W}_0^\mathtt{P} + \mathbf{x}(\mathbf{B}^\mathtt{P}\mathbf{A}^\mathtt{P}). \tag{4}$$

**Recovered Low-Rank Matrix Generation.** By optimizing the objective function $\mathcal{L}_{\text{SFT}}$, we obtain the trained pruned low-rank matrix $\mathbf{W}_\Delta^{\mathtt{P}^\star}$. To fully leverage the original model weights for improved inference performance, we introduce a recovery function $\mathtt{R}(\cdot)$, guided by the pruning mask $\mathbf{M}^\mathtt{P}$. This function recovers the shape of the trained low-rank matrix by filling zeros at pruned positions, resulting in $\mathbf{W}_\Delta^{\mathtt{R}^\star}$ as follows:

$$\mathbf{W}_\Delta^{\mathtt{R}^\star} = \mathbf{B}^{\mathtt{R}^\star}\mathbf{A}^{\mathtt{R}^\star} = \mathtt{R}(\mathbf{W}_\Delta^{\mathtt{P}^\star}) = \mathbf{W}_\Delta^{\mathtt{P}^\star} \circ (1 - \mathbf{M}^\mathtt{P}). \tag{5}$$

This operation ensures that the recovered low-rank matrix $\mathbf{W}_\Delta^{\mathtt{R}^\star}$ can be seamlessly merged with the original pre-trained weights $\mathbf{W}_0$, forming $\mathbf{W}_0 + \mathbf{W}_\Delta^{\mathtt{R}^\star}$ as follows:

$$(\mathbf{W}_0 + \mathbf{W}_\Delta^{\mathtt{R}^\star})[i,j] = \begin{cases} \mathbf{W}_0[i,j] & \text{if } \mathbf{M}^\mathtt{P}[i,j] = 1 \\ \mathbf{W}_0[i,j] + \mathbf{W}_\Delta^{\mathtt{R}^\star}[i,j] & \text{if } \mathbf{M}^\mathtt{P}[i,j] = 0 \end{cases} \tag{6}$$

This formula indicates that, for positions where the pruning mask $\mathbf{M}^\mathtt{P}$ is '1', the merged matrix $\mathbf{W}_0 + \mathbf{W}_\Delta^{\mathtt{R}^\star}$ retains the original values from the pre-trained matrix $\mathbf{W}_0$. For positions where the mask is '0', the elements in the merged matrix are updated to be the sum of the corresponding values from $\mathbf{W}_0$ and the recovered low-rank matrix $\mathbf{W}_\Delta^{\mathtt{R}^\star}$.

**Recovered Low-Rank Matrix Inference.** Once we obtain the recovered low-rank matrix $\mathbf{W}_\Delta^{\mathtt{R}^\star}$, during inference, the forward pass output activation $\mathbf{h}$ for an input feature $\mathbf{x}$ is computed as follows:

$$\mathbf{h} = \mathbf{x}(\mathbf{W}_0 + \mathbf{W}_\Delta^{\mathtt{R}^\star}) = \mathbf{x}(\mathbf{W}_0 + \mathbf{B}^{\mathtt{R}^\star}\mathbf{A}^{\mathtt{R}^\star}). \tag{7}$$

Our experiments (see Section 3.2 and Section 3.3) show that LoRAM maintains high performance across various pruning strategies $\mathtt{P}(\cdot)$, including structured (Ma et al., 2023) and non-structured pruning (semi-structured & unstructured pruning) (Frantar & Alistarh, 2023a). The four stages outlined above summarize the core steps of LoRAM. To avoid notational clutter, we have streamlined the algorithmic details. Nonetheless, three key considerations for deployment must be emphasized:

$C_1$ **Pruned Full-Rank Weight Generation:** For non-structured pruning, the matrix dimension remains unchanged, with $\mathbf{W}_0^P$ compressed into a sparse matrix populated by zeros. In structured pruning, weights are physically removed, yielding a compact, dense $\mathbf{W}_0^P$.

$C_2$ **Recovered Low-Rank Matrix Generation:** For non-structured pruning, the weights in $\mathbf{W}_\Delta^P$ corresponding to pruned positions in $\mathbf{M}^P$ are excluded from backpropagation by setting their gradients to zero, ensuring that only the retained components are updated during training.

$C_3$ **Recovered Low-Rank Matrix Inference:** For non-structured pruning, the shapes of both the pre-trained and low-rank weights are identical (see $C_1$), and the gradients of the pruned weights are blocked (see $C_2$). Consequently, we can bypass the recovery phase, resulting in $\mathbf{W}_\Delta^{R^\star} = \mathbf{B}^{R^\star}\mathbf{A}^{R^\star} = \mathbf{B}^{P^\star}\mathbf{A}^{P^\star}$. In the case of structured pruning, the shapes of the weight matrices vary significantly across different pruning strategies. To simplify the definitions, we standardize the recovery process using the pruning mask.

To clearly illustrate the evolution of weight matrix dimensions across these stages, we take LLM-Pruner (Ma et al., 2023) as an example in Appendix C, visualizing the transformation from $\mathbf{W}_0 \Rightarrow \mathbf{W}_0^P$, $\mathbf{W}_\Delta \Rightarrow \mathbf{W}_\Delta^P$, and $\mathbf{W}_\Delta^{P^\star} \Rightarrow \mathbf{W}_\Delta^{R^\star}$ under LoRAM with structured pruning.

**Pruned Full-Rank Weight Alignment.** Given the original pre-trained weights $\mathbf{W}_0$, the optimal low-rank matrix learned is $\mathbf{W}_\Delta^\star$. Similarly, for pruned weights $\mathbf{W}_0^P$, the optimal low-rank counterpart is $\mathbf{W}_\Delta^{P^\star}$. If the knowledge encoded in $\mathbf{W}_0$ and $\mathbf{W}_0^P$ is closely aligned, the knowledge embedded in $\mathbf{W}_\Delta^{P^\star}$ should approximate that of $\mathbf{W}_\Delta^\star$. Consequently, the recovered matrix $\mathbf{W}_\Delta^{R^\star}$ should effectively pair with $\mathbf{W}_0$, yielding performance improvements similar to those from $\mathbf{W}_\Delta^\star$. However, the pruning function $P(\cdot)$ disrupts some of the knowledge embedded in the original weights, leading to a mismatch between $\mathbf{W}_0$ and $\mathbf{W}_0^P$. Such knowledge mismatch causes $\mathbf{W}_\Delta^{R^\star}$, when paired with $\mathbf{W}_0$, to deliver suboptimal performance, particularly at higher pruning ratios.

To address the knowledge mismatch, we propose an efficient alignment scheme, namely continual pre-training of pruned weights $\mathbf{W}_0^P$ into $\mathbf{W}_0^{P,A}$ on a small, general corpus $\mathcal{D}_A$. Formally, we minimize the alignment loss $\mathcal{L}_A$ defined as following:

$$\mathcal{L}_A = -\mathbb{E}_{\mathbf{s} \in \mathcal{D}_A} \left[ \sum_{t=1}^{|\mathbf{s}|} \log p\left(\mathbf{s}_{t+1} \mid \mathbf{s}_{<t}; \mathbf{W}_0^{P,A}\right) \right], \tag{8}$$

where $p(\mathbf{s}_{t+1} \mid \mathbf{s}_{<t}; \mathbf{W}_0^{P,A})$ represents the model's predict likelihood of generating the next token $\mathbf{s}_{t+1}$ given the first $t$ tokens $\mathbf{s}_{<t}$ of input sequence $\mathbf{s}$ (token number is $|\mathbf{s}|$) and current parameter matrices $\mathbf{W}_0^{P,A}$. This alignment process is a one-time, offline operation on a relatively small corpus (about 105 million tokens), making it a cost-effective solution for model publishers. Alongside the base model, they can release the aligned pruned model, enabling low-resource users to fine-tune the base model via LoRAM for specific downstream tasks.

**Pruned Full-Rank Weight Quantization.** The design of LoRAM inherently supports the seamless integration of quantization techniques, further reducing memory consumption during training by applying quantization to pruned models. For example, by adapting the LoRA-specific quantization scheme QLoRA (Dettmers et al., 2023), LoRAM extends into QLoRAM, the pruned full-rank weight matrix is quantized into the NF4 format, while the pruned low-rank matrices $\mathbf{B}^P$ and $\mathbf{A}^P$ remain in full or half precision, striking a balance between memory efficiency and fine-tuning quality.

Formally, given a quantization function $Q(\cdot)$, during training, the forward pass output activation vector $\mathbf{h}$ for an input feature vector $\mathbf{x}$ is computed as:

$$\mathbf{h} = \mathbf{x}Q(\mathbf{W}_0^P) + \mathbf{x}\mathbf{B}^P\mathbf{A}^P = \mathbf{x}\mathbf{W}_0^{P,Q} + \mathbf{x}\mathbf{B}^P\mathbf{A}^P, \tag{9}$$

where $\mathbf{W}_0^{P,Q}$ represents the full-rank weight $\mathbf{W}_0$ after undergoing pruning via $P(\cdot)$, followed by quantization using $Q(\cdot)$. We can also quantize the pruned full-rank weight after alignment to $\mathbf{W}_0^{P,A,Q}$. For inference, unless additional quantization is required, QLoRAM operates identically to LoRAM, as shown in Fig. 2 (d). It utilizes the original full-rank weights $\mathbf{W}_0$ alongside the recovered low-rank matrices $\mathbf{B}^{R^\star}$ and $\mathbf{A}^{R^\star}$ to perform the forward pass according to Eq. (7). In summary, for

LoRAM, as shown in Fig. 2 (e), the offline processing path of the frozen full-rank matrix $\mathbf{W}_0^*$, which minimizes GPU memory usage, is $\mathbf{W}_0 \xrightarrow{\text{P}(\cdot)} \mathbf{W}_0^{\text{P}} \xrightarrow{\mathcal{L}_A} \mathbf{W}_0^{\text{P,A}} \xrightarrow{\text{Q}(\cdot)} \mathbf{W}_0^{\text{P,A,Q}}$; Fig. 2 (f) shows that the online generation path for the trained low-rank matrix $\mathbf{W}_\Delta^*$ is $\mathbf{W}_\Delta \xrightarrow{\text{P}(\cdot)} \mathbf{W}_\Delta^{\text{P}} \xrightarrow{\mathcal{L}_{\text{SFT}}} \mathbf{W}_\Delta^{\text{P}\star} \xrightarrow{\text{Q}(\cdot)} \mathbf{W}_\Delta^{\text{R}\star}$.

## 3 EXPERIMENTS

### 3.1 SETUP

**Pre-train Corpus.** To align the inconsistent knowledge between the pruned model during training and the original model during inference, we apply LoRAM to continual pre-training LLMs on a mixed corpus of FineWeb (Penedo et al., 2024) and OpenWebMath (Paster et al., 2023). Notably, this alignment process is a one-time, offline operation that can be executed by model publishers.

**Fine-tuning Data.** Following the fine-tuning setup of LoRA (Hu et al., 2022), we primarily conduct supervised fine-tuning (SFT) on the OpenHermes-2.5 (Teknium, 2023) (referred to as Open-Hermes) and OpenOrca (Lian et al., 2023) datasets. To effectively assess the overall fine-tuning performance, we evaluate test perplexity not only on in-domain test sets constructed from the instruction fine-tuning data but also on out-of-domain test sets built from Alpaca (Taori et al., 2023).

**Downstream Task.** We focus on the performance of LoRAM in various downstream tasks, including MathQA (Amini et al., 2019) and GSM8K (Cobbe et al., 2021) in mathematical reasoning, six tasks—Arc Challenge & Easy (Clark et al., 2018), HellaSwag (Zellers et al., 2019), Open-BookQA (Mihaylov et al., 2018), PIQA (Bisk et al., 2020), and WinoGrande (Sakaguchi et al., 2021) in common sense reasoning, and HumanEval (Chen et al., 2021) in code generation.

**Sparsification & Quantization.** For sparsification $\text{P}(\cdot)$, we first establish a variant LoRAM-RAND randomly structured pruning and adapt LoRAM to another three variants based on leading approaches: LoRAM-STRU with the structured pruning LLM-Pruner[2] (Ma et al., 2023) and LoRAM-SEMI and LoRAM-UNST with the non-structured (semi-structured & unstructured) pruning SparseGPT[3] (Frantar & Alistarh, 2023a). For quantization $\text{Q}(\cdot)$, we achieve QLoRAM by combining LoRAM with the LoRA-tailored quantization algorithm QLoRA (Dettmers et al., 2023). The storage cost of the original model primarily drives the memory consumption during LoRA weights training. Thus, we define the *parameter reduction ratio* as the count of parameters before and after pruning, to evaluate the memory efficiency of baselines. The details of our experiment setups and hyperparameters are provided in Appendix B.

### 3.2 FINE-TUNING CONVERGENCE

We investigate the convergence trends of LoRAM across varying model scales (LLaMA-2-13B & LLaMA-2-70B) and different instruction-tuning datasets (OpenHermes & OpenOrca). To assess training performance, we track perplexity over training iterations on both out-of-domain (Alpaca) and in-domain (OpenHermes or OpenOrca) test sets, as shown in Fig. 3 and Fig. 4.

**Out-of-Domain Performance.** LoRAM consistently achieves out-of-domain performance with similar trends, positioned between the LoRA fine-tuned models of the same scale and smaller models, across different models and datasets. As shown in Figs. 3 and 4 (a), for the 13B model, the perplexity of LoRAM variants pruned by different algorithms is lower than that of the LoRA-trained 7B model but higher than the LoRA-trained 13B model, with LoRAM-RAND and LoRAM-STRU achieving a 2.17× parameter reduction. Similarly, as shown Figs. 3 and 4 (c), for the 70B model, this reduction extends to 12.84× under similar convergence trends.

**In-Domain Performance.** LoRAM shows limited improvement in in-domain performance, likely due to overfitting when the base models are fine-tuned with LoRA, resulting in relatively lower

---

[2] https://github.com/horseee/LLM-Pruner (Apache-2.0 license)
[3] https://github.com/IST-DASLab/sparsegpt (Apache-2.0 license)

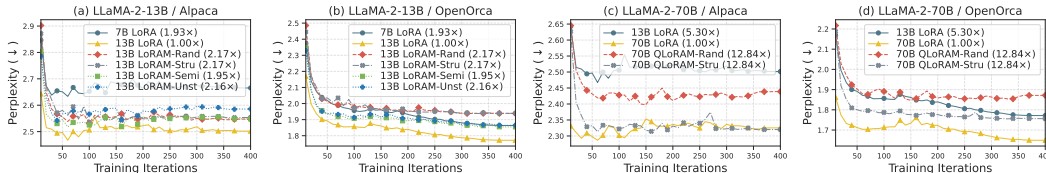

Figure 3: The test perplexity of training LLaMA-2-13B & LLaMA-2-70B on OpenHermes.

Figure 4: The test perplexity of training LLaMA-2-13B & LLaMA-2-70B on OpenOrca.

perplexity. This is further supported by downstream evaluations, where models that excel in in-domain perplexity often underperform in downstream tasks. As shown in Figs. 3 and 4 (b), while the LoRA-trained 7B model outperforms 13B LoRAM-Rand and LoRAM-Stru on in-domain tests, it underperforms on several downstream tasks as shown in Section 3.3.

**Non-Structured LoRAM Excels in In-Domain.** The non-structured variants (LoRAM-Semi & LoRAM-Unst) consistently outperform their structured counterparts (LoRAM-Rand & LoRAM-Stru) on in-domain test sets. As shown in Fig. 3 (a) vs. Fig. 3 (b) and Fig. 4 (a) vs. Fig. 4 (b), the in-domain perplexity of LoRAM-Semi and LoRAM-Unst is notably lower, while their out-of-domain performance shows less pronounced differences. This advantage likely arises from the more selective weight pruning in the non-structured variants, which preserves information capture capabilities similar to the original model, thus enhancing in-domain performance.

**Non-Random LoRAM Benefits from Scaling.** The performance gains of the non-random LoRAM become more evident as the model size grows. As shown in (a,b) of Figs. 3 and 4 vs. (c,d) of Figs. 3 and 4, LoRAM-Stru outperforms LoRAM-Rand considerably on the 70B model, while the difference is marginal on the 13B model. This indicates that larger models exhibit greater differences in the redundancy of individual weights, making selective pruning more effective[4].

### 3.3 DOWNSTREAM TASK PERFORMANCE

We evaluate the performance of various models trained with LoRAM on different instruction data across three downstream tasks: mathematical reasoning, common sense reasoning (CSR), and code generation. Results are summarized in Tables 1 to 3. We highlight the *core competition scenario* with a gray background, which includes the untrained original model and a smaller sibling model trained with LoRA. For instance, for LoRAM-trained LLaMA-2-13B, we report the scores of the 13B without fine-tuning and the LoRA-trained 7B model. Blue backgrounds of varying intensity indicate the degree of improvement for each LoRAM variant relative to the *core competition scenario*: darker shades indicate greater improvements, while lighter shades signify smaller gains.

Overall, we observe that most LoRAM variants outperform the core competitive baseline across all downstream tasks, particularly in mathematical and common sense reasoning. This improvement is further amplified by increasing the model scale. Specifically, as shown in Table 1, the 70B LoRAM-Rand and LoRAM-Stru models achieve a 12.84× reduction in parameters compared to the original 70B model (70B w/o FT), exceeding the 5.30× reduction of the LoRA-trained 13B model. In terms of performance, LoRAM improves the original 70B model's score on GSM8K from 52% to 57%, significantly outperforming the LoRA-trained 13B model, which only achieved 37%. These results demonstrate that updating low-rank matrices on pruned models effectively re-

---

[4]The trained low-rank matrices are visualized in Appendix D, and the update patterns they exhibit somewhat align with these insights.

Table 1: Accuracy (%) of the MathQA (1-shot) & GSM8K (8-shots) in the mathematical domain under LLaMA-2. ▲ indicates the theoretical parameters reduction of non-structured pruning. However, these parameters are filled with zeros in actual training, so the memory footprint is not reduced.

| Method | OpenHermes | | OpenOrca | | Parameter |
|---|---|---|---|---|---|
| | MathQA | GSM8K | MathQA | GSM8K | Redu. Ratio |
| 13B w/o FT | 32.60 | 24.26 | 32.93 | 23.35 | 1.00× |
| 7B LoRA | 29.61 | 22.82 | 30.95 | 13.87 | 1.93× |
| 13B LoRAM-Rand | 33.77 | 27.22 | 32.83 | 25.93 | 2.17× |
| 13B LoRAM-Stru | 33.80 | 24.64 | 33.07 | 24.49 | 2.17× |
| 13B LoRAM-Semi | 31.76 | 36.92 | 33.07 | 27.29 | ▲ 1.95× |
| 13B LoRAM-Unst | 30.12 | 31.92 | 32.70 | 26.61 | ▲ 2.16× |
| 70B w/o FT | 39.53 | 52.01 | 39.53 | 52.01 | 1.00× |
| 13B LoRA | 32.03 | 36.69 | 33.63 | 25.70 | 5.30× |
| 70B QLoRAM-Rand | 39.66 | 57.62 | 39.40 | 55.72 | 12.84× |
| 70B QLoRAM-Stru | 39.77 | 57.16 | 39.73 | 54.44 | 12.84× |

Table 2: Average accuracy (%) of the CSR in the common sense reasoning domain (1-shot) under the LLaMA-2. Baseline results for each subtask of CSR are detailed in Appendix E.

| Method | OpenHermes | OpenOrca | Parameter |
|---|---|---|---|
| | Mean ± Std | Mean ± Std | Redu. Ratio |
| 13B w/o FT | 64.28±1.30 | 64.28±1.30 | 1.00× |
| 7B LoRA | 61.51±1.29 | 61.42±1.30 | 1.93× |
| 13B LoRAM-Rand | 64.64±1.29 | 64.49±1.30 | 2.17× |
| 13B LoRAM-Stru | 64.42±1.29 | 64.32±1.29 | 2.17× |
| 13B LoRAM-Semi | 64.38±1.29 | 64.73±1.30 | ▲ 1.95× |
| 13B LoRAM-Unst | 64.12±1.29 | 64.68±1.29 | ▲ 2.16× |
| 70B w/o FT | 68.69±1.27 | 68.69±1.27 | 1.00× |
| 13B LoRA | 65.05±1.29 | 65.40±1.29 | 5.30× |
| 70B QLoRAM-Rand | 68.99±1.27 | 68.46±1.27 | 12.84× |
| 70B QLoRAM-Stru | 69.10±1.27 | 68.94±1.27 | 12.84× |

Table 3: Pass@1(%) and Pass@10(%) of HumanEval in the code generation domain under LLaMA-2. The best results for all baselines are reported, selected from Temperature settings in {0.0, 0.2, 0.4, 0.6, 0.8} with Top_P fixed at 0.95.

| Method | OpenHermes | | OpenOrca | | Parameter |
|---|---|---|---|---|---|
| | Pass@1 | Pass@10 | Pass@1 | Pass@10 | Redu. Ratio |
| 13B w/o FT | 17.68 | 35.37 | 17.68 | 35.37 | 1.00× |
| 7B LoRA | 15.24 | 28.04 | 15.85 | 26.21 | 1.93× |
| 13B LoRAM-Rand | 19.51 | 33.54 | 19.51 | 32.32 | 2.17× |
| 13B LoRAM-Stru | 17.68 | 35.37 | 17.07 | 31.71 | 2.17× |
| 13B LoRAM-Semi | 20.12 | 35.37 | 18.29 | 39.63 | ▲ 1.95× |
| 13B LoRAM-Unst | 22.56 | 34.15 | 18.29 | 37.20 | ▲ 2.16× |
| 70B w/o FT | 31.71 | 58.54 | 31.71 | 58.54 | 1.00× |
| 13B LoRA | 18.29 | 35.98 | 18.29 | 39.02 | 5.30× |
| 70B QLoRAM-Rand | 29.27 | 57.32 | 31.71 | 56.71 | 12.84× |
| 70B QLoRAM-Stru | 32.32 | 58.54 | 32.32 | 59.15 | 12.84× |

duces memory requirements during training. Merging the recovered low-rank matrices into the original model yields substantial performance gains during inference.

## 3.4 Adaption to LLaMA-3.1

Here, we extend LoRAM-Stru to LLaMA-3.1 herds and investigate two key questions: (1) How does LoRAM perform in terms of perplexity and downstream tasks within this model series? (2) What is the effect of continued pre-training iteration steps (proportional to corpus size) on performance?

As shown in Fig. 5 (a,b), QLoRAM-Stru for the LLaMA-3.1-70B model exhibits consistent trends across both out-of-domain and in-domain test sets. It achieves a 15.81× parameters reduction while its perplexity falls between that of the smaller LoRA-trained 8B and LoRA-trained 70B.

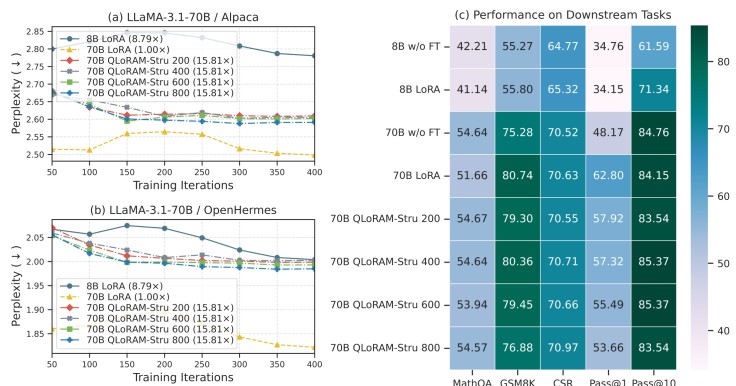

Figure 6: Necessity of Recovery & Alignment across different pruning strategies on LLaMA-2-13B.

In downstream tasks (Fig. 5 (c)), QLoRAM-STRU significantly exceeds the 8B w/o FT, 8B LoRA, and 70B w/o FT, even surpassing the LoRA-trained 70B on MathQA and HumanEval (Pass@10). Moreover, we observe that a minimal pre-training corpus can yield substantial performance gains. For instance, QLoRAM-STRU 200, with just 200 updates (about 13 million tokens), achieves a 15.81× parameter reduction alongside performance improvements. This one-time, low-cost alignment allows publishers to offer aligned pruned models for low-resource users to customize tasks.

Figure 5: The test perplexity & downstream performance of training LLaMA-3.1-70B on OpenHermes.

## 3.5 NECESSITY OF RECOVERY & ALIGNMENT

We conduct an ablation study on two critical phases of LoRAM: recovery and alignment. To assess their necessity, we analyze the convergence trends of various pruning variants on the Alpaca test set using LLaMA-2-13B.

**Impact of Recovery.** we compare the standard approach with an alternative setup where the pruned low-rank matrices are directly combined with the pruned full-rank model weights (w/o Recovery) and track perplexity changes over iterations. As shown in Fig. 6, for all four pruning strategies, models without the recovery phase (solid lines, w/o Recovery & *) consistently exhibit higher perplexity compared to those with recovery (dashed lines, w/ Recovery & *), particularly in structured LoRAM (see in Fig. 6 (a) and (b)). This highlights that the recovery phase leverages relatively redundant neurons during training to enhance inference performance significantly.

**Impact of Alignment.** We also introduce a variant of LoRAM without continual pre-training for alignment (w/o Alignment). As shown in Fig. 6, aligned pruned models (yellow lines, * & w/ Alignment) consistently achieve lower perplexity than unaligned counterparts (blue lines, * & w/o Alignment), irrespective of the pruning strategy or recovery phase. This highlights that even low-cost continual pre-training on a small general corpus effectively narrows the knowledge gap between pruned and original models, enhancing the overall performance of LoRAM.

## 3.6 SCALING LAWS FOR PARAMETER REDUCTION ON LoRAM

We explore the impact of scaling the parameter reduction ratios in Fig. 7. The LoRA-trained LLaMA-2-13B (triangles) achieves a 5.30× parameter reduction, while QLoRAM-STRU maintains superior perplexity on the Alpaca and further reduces parameters across both instruction datasets. In contrast, naive pruning leads to a significant increase in perplexity with minimal pruning. When the parameter reduction ratio reaches 28.56×, QLoRAM-STRU sustains an effective perplexity of approximately 2.5, whereas naive pruning escalates to 621.98. These highlight LoRAM's ability to

drastically reduce memory of the base model by updating LoRA weights in the pruned model, while seamlessly integrating with the full model to preserve inference performance.

We then evaluate the performance of models trained with LoRAM on OpenHermes across various downstream tasks under different pruning ratios. As shown in Fig. 8, overall performance improves as the parameter reduction ratio increases from 9.82× to 16.95×, before declining. Notably, tasks achieve optimal performance between parameter reduction ratios of 12.84× and 16.95×, consistently outperforming 13B LoRA and 70B w/o FT. However, at a parameter reduction ratio of 9.82×, despite the larger memory capacity available for LoRAM, downstream performance does not always exceed that of higher parameter reduction ratios. We attribute this to the fact

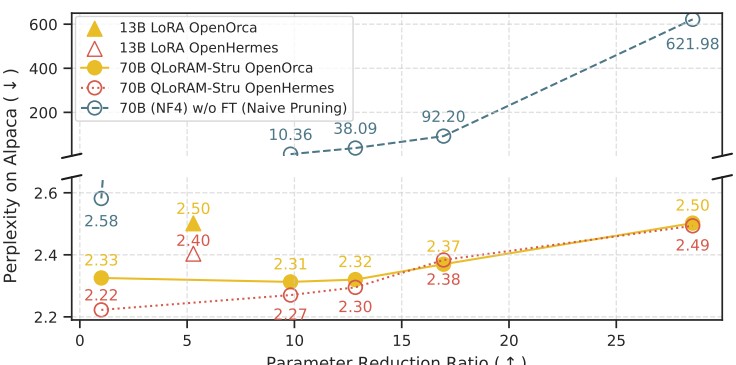

Figure 7: Effect of scaling parameter reduction ratio.

that lower parameter reduction ratios fine-tune more parameters, potentially degrading the pre-trained model's performance on certain tasks (e.g., Fig. 8 (a,c,e)). This effect is also reflected in MathQA, where a fully fine-tuned LoRA model underperforms the pre-trained model without fine-tuning (see Fig. 8 (b)). Moreover, excessive pruning at a ratio of 28.56× leaves too few neurons to capture the rich information needed for downstream improvements, particularly in tasks like code generation (see Fig. 8 (d,e)).

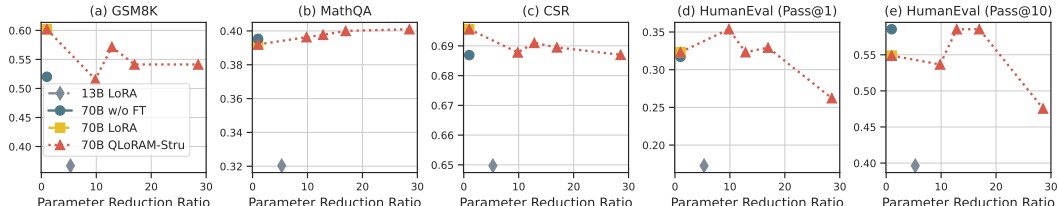

Figure 8: Performance of downstream tasks across different parameter reduction ratios.

## 4 CONCLUSION

We propose LoRAM, a memory-efficient LoRA training scheme for large language models. LoRAM significantly reduces the count of parameters of the original model by 16.95×, while maintaining the performance of large-scale LLM fine-tuning. We identify several open questions for LoRAM, including the potential for reduced inference costs through context-aware computational graph recovery and its applicability to models like vision transformers (Dosovitskiy et al., 2021) and diffusion models (Ho et al., 2020). We hope our work inspires further research on memory-efficient LoRA training from a sparsity perspective and believe LoRAM will serve as a valuable tool for the community, enabling LoRA training of large-scale models on consumer-grade hardware.

ACKNOWLEDGMENTS

This work is supported by the Pioneer R&D Program of Zhejiang (No. 2024C01021), "Leading Talent of Technological Innovation Program" of Zhejiang Province (No. 2023R5214), OPPO Research Fund, the Major Research Program of Zhejiang Provincial Natural Science Foundation (No. LD24F020015), and NSFC Grant No. 62402420.

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

APPENDIX

# A  RELATED WORK

**Low-Rank Adaptation.**  LoRA (Low-Rank Adaptation) (Hu et al., 2022) has emerged as a prominent technique for parameter-efficient fine-tuning (PEFT) (Li & Liang, 2021; Lester et al., 2021; Liu et al., 2021; Qiu et al., 2023; Liu et al., 2024b; 2022). By injecting lightweight, trainable low-rank decomposition matrices into frozen pre-trained weights, LoRA enables efficient task customization, especially in resource-constrained settings. Some LoRA variants (Liu et al., 2024a; Ding et al., 2023; Zi et al., 2023; Zhang et al., 2023b; Kalajdzievski, 2023) have been developed to enhance its generalization and robustness, while others (Zhou et al., 2024; Zhang et al., 2023a; Kopiczko et al., 2024; Azizi et al., 2024; Wang et al., 2024; Yang et al., 2024) address the increased memory overhead associated with scaling up model sizes. However, during training, these efficient LoRA variants still struggle with the substantial memory footprint of the original LLM parameters.

**LoRA-related Compression.**  Model compression techniques like quantization (Han et al., 2015; Jacob et al., 2018; Nagel et al., 2019; Zhao et al., 2019; Yao et al., 2022; Park et al., 2022; Dettmers et al., 2022; Xiao et al., 2022; Frantar et al., 2022), sparsification (Molchanov et al., 2016; Liu et al., 2018; He et al., 2019; Hoefler et al., 2021; Frantar & Alistarh, 2023b; Liu et al., 2023; Bansal et al., 2022), and distillation (Hinton et al., 2015; Cho & Hariharan, 2019; Tang et al., 2019; Touvron et al., 2021; Hsieh et al., 2023; Gu et al., 2024b) have proven effective in reducing the memory footprint of LLM during training and inference. Naturally, the concept of compression has been adapted to LoRA to alleviate the substantial memory consumption dominated by pre-trained model parameters. In particular, LoRA-related quantization schemes (Dettmers et al., 2023; Xu et al., 2024; Li et al., 2024; Guo et al., 2024; Frantar et al., 2023; Chai et al., 2023) have been widely explored, but they still face the limitations of 1-bit precision, typically quantize weights to 4-bit to balance training efficiency with performance. Our work aims to push the boundaries of memory-efficient LoRA training by leveraging sparsification to achieve cost-effective performance improvements. Notably, existing LoRA-related sparsification works (Chen et al., 2023; Zhang et al., 2024a) focus on designing pruning algorithms to slim down models and use LoRA to recover the knowledge of pruned models, thereby producing compact but high-quality models. In contrast, LoRAM enables effective general pruning under high base-model sparsity, whereas (Gu et al., 2024a) focuses on task-specific LoRA sparsification with limited impact on base model memory reduction.

# B  EXPERIMENTAL DETAILS

**Pre-train Corpus.**  To align the inconsistent knowledge between the pruned model during training and the original model during inference, we apply LoRAM to continual pre-training LLMs in a teacher-forcing manner (Bachmann & Nagarajan, 2024) on a mixed corpus of FineWeb (Penedo et al., 2024) and OpenWebMath (Paster et al., 2023). FineWeb, containing over 15TB of cleaned and deduplicated English web data from Common Crawl. OpenWebMath, extracted from over 200 billion HTML files on Common Crawl, provides high-quality mathematical text. Mixing these datasets enhances the pruned model's capabilities in both general and mathematical domains.

Unless specified otherwise, we randomly sample 102,400 instances from both FineWeb and OpenWebMath to construct a mixed dataset with a sequence length of 512, yielding approximately 105 million tokens. The default training batch size is 128, allowing up to 1,600 update steps. We train without data repetition over a sufficiently large corpus to simulate a realistic pre-training scenario. Notably, this alignment process is a one-time, offline operation that model publishers can execute.

**Fine-tuning Data.**  Following the fine-tuning scenario of LoRA (Hu et al., 2022), we primarily conduct supervised fine-tuning (SFT) on the OpenHermes-2.5 (Teknium, 2023) (referred to as Open-Hermes). OpenHermes is a large-scale dataset constructed from synthetically generated instructions and chat samples, encompassing diverse sources such as Airoboros 2.2 (Wang et al., 2023), CamelAI Domain Expert Dataset (Li et al., 2023a), ChatBot Arena (GPT-4 Only) (Zheng et al., 2023), and more. To further demonstrate the general effectiveness of the LoRAM alignment process, we also evaluate LoRAM on the OpenOrca (Lian et al., 2023) dataset. OpenOrca is a widely used

instruction fine-tuning dataset where each data instance represents entries from the FLAN collection (Longpre et al., 2023), augmented by submitting the listed questions to either GPT-4 or GPT-3.5.

By default, we train SFT on the instruction dataset with a batch size of 128 and a sequence length of 512 for 400 steps, totaling approximately 26.2 million tokens. To effectively evaluate the overall fine-tuning performance, we assess the perplexity of the fine-tuned model on an out-of-domain test set. This out-of-domain test set is constructed by randomly sampling 2,000 instances from the Alpaca (Taori et al., 2023) test set, truncated to a sequence length of 512.

**Downstream Task.** We focus on the performance of LORAM in various downstream tasks, including mathematical reasoning, common sense reasoning, and code generation. All our downstream task evaluations are performed on lm-evaluation-harness[5] and code-eval [6] with VLLM [7].

For mathematical reasoning, we benchmark the accuracy of baseline models using greedy decoding on MathQA (Amini et al., 2019) with a 1-shot setting and GSM8K (Grade School Math 8K) (Cobbe et al., 2021) with 8-shots, Chain of Thought (CoT) prompting and strict match MathQA is a large-scale dataset comprising 37k English multiple-choice math word problems, covering diverse math domains. It extends the AQuA-RAT dataset (Ling et al., 2017) by annotating problems with fully specified operational programs using a new representation language, building on the questions, options, rationale, and correct answers provided by AQuA-RAT. The GSM8K is a dataset of 8.5K high-quality, linguistically diverse grade school math word problems, designed to evaluate multi-step reasoning in basic arithmetic operations (+-×÷). We conduct evaluations on its 1.3K test set with *strict-match* to assess logical and mathematical reasoning in language models.

For commonsense reasoning (CSR), we report the average accuracy across six tasks—Arc Challenge & Easy (Clark et al., 2018), HellaSwag (Zellers et al., 2019), OpenBookQA (Mihaylov et al., 2018), PIQA (Bisk et al., 2020), and WinoGrande (Sakaguchi et al., 2021)—under 1-shot and greedy decoding settings. These benchmarks comprehensively assess the model's ability to apply "commonsense" or world knowledge for reasoning, rather than relying on pattern recognition.

For code generation, we compare two pass rates, PASS@1 and PASS@10 (Kulal et al., 2019), on HumanEval (Chen et al., 2021) of each baseline in a zero-shot setting with sampling parameters of TEMPERATURE $= \{0.0, 0.2, 0.4, 0.6, 0.8\}$, and TOP$_P = 0.95$. The HumanEval dataset released by OpenAI consists of 164 handwritten Python programming problems, each with a function signature, docstring, body, and unit tests. Serving as a benchmark, HumanEval assesses models on a range of Python coding skills, from basic syntax to complex problem-solving, offering insights into their programming capabilities alongside language-focused tasks.

**Sparsification & Quantization.** LORAM incorporates two model compression techniques: sparsification, which generates a pruned model for low-rank matrix updates, and quantization, which forms QLORAM further to reduce the memory footprint of the pruned model. For sparsification, to validate the general effectiveness of LORAM, we benchmark its performance across various pruning strategies P($\cdot$). Specifically, we first establish a variant using randomly structured pruning and adapt LORAM to another three variants based on leading approaches: the structured pruning LLM-Pruner[8] (Ma et al., 2023) and the non-structured (semi-structured & unstructured) pruning SparseGPT[9] (Frantar & Alistarh, 2023a). These baselines are summarized below, with the corresponding configurations presented in Tables 4 to 6.

- **LORAM-RAND**: We adhere to the pruning settings of LORAM-STRU, modifying only by randomly removing weights instead of the original gradient-based pruning criterion.

- **LORAM-STRU**: We follow LLM-Pruner and employ a block-wise strategy for local structured pruning. Attention and MLP layers are treated as separate blocks, with non-critical coupling weights pruned based on gradient information at a uniform ratio. We retain the first four and last two layers of both blocks, focusing pruning on the intermediate layers.

---

[5] https://github.com/EleutherAI/lm-evaluation-harness (MIT License).
[6] https://github.com/abacaj/code-eval (MIT License).
[7] https://github.com/vllm-project/vllm (Apache-2.0 license).
[8] https://github.com/horseee/LLM-Pruner (Apache-2.0 license)
[9] https://github.com/IST-DASLab/sparsegpt (Apache-2.0 license)

- **LORAM-SEMI**: We utilize SparseGPT with a 4:8 semi-structured sparsity pattern to prune pre-trained weights across all model layers.

- **LORAM-UNST**: We prune individual weights uniformly across layers using a predefined pruning ratio based on an unstructured version of SparseGPT.

For quantization $Q(\cdot)$, to further reduce memory usage during training, especially when dealing with models exceeding 70 billion parameters, we achieve QLORAM by combining LORAM with the LoRA-tailored quantization algorithm QLoRA (Dettmers et al., 2023). While LORAM is compatible with the quantization of other customized LoRA methods (Xu et al., 2024; Li et al., 2024; Guo et al., 2024; Frantar et al., 2023; Chai et al., 2023), this falls outside the scope of this article.

**Architecture & Hyperparameters.** We adopt a LLaMA architecture with RMSNorm (Zhang & Sennrich, 2019) and SwiGLU activations (Shazeer, 2020; Zhao et al., 2022). We run all experiments with BF16 format to reduce memory usage. For all configurations, we default to a learning rate of 1e-3. However, the downstream performance of models fine-tuned on OpenOrca is relatively sensitive to the learning rate. Therefore, in this evaluation, we tune the learning rates for each baseline within the range of [1e-5, 1e-3] and report their respective optimal downstream scores. Specifically, we use 1e-5 for the 7B LoRA and 13B & 70B LoRAM models, and 1e-4 for the 13B LoRA model. All experiments run on NVIDIA A100-80GB GPUs with environments of CUDA 12.2, PyTorch 2.4.0, and Transformer 4.45.1. For LLaMA-2 herds, we set low-rank matrices $\mathbf{B}$ and $\mathbf{A}$ of rank $r = 8$ for $\mathbf{W}_q$, $\mathbf{W}_k$, $\mathbf{W}_v$, and $\mathbf{W}_o$ in the attention layer, $\mathbf{W}_{up}$, $\mathbf{W}_{gate}$, and $\mathbf{W}_{down}$ in the MLP layer, and the head embedding matrix $\mathbf{W}_{lm\_head}$; for LLaMA-3 herds, we exclude the injection of the low-rank matrix of $\mathbf{W}_{lm\_head}$.

Table 4: LoRAM configures on LLaMA-2-13B. Comparison of different pruning methods in terms of parameter reduction ratio (Reduction) and HBM footprint (GB) of pruned parameters (HBM), ignoring low-rank matrix overhead.

| Method | #Orig. Params | Pruning Ratio | #Pruned Params | Reduction | HBM |
|---|---|---|---|---|---|
| LoRAM-Semi | 13015864320 | 0.50 | 6738415616 | 1.93× | 12.55 |
| LoRAM-Unst | 13015864320 | 0.55 | 6037628912 | 2.16× | 11.25 |
| LoRAM-Rand & Stru | 13015864320 | 0.65 | 6005662720 | 2.17× | 11.19 |

Table 5: LoRAM configures on LLaMA-2-70B and LLaMA-3.1-70B with different pruning ratios.

| Method | #Orig. Params | Pruning Ratio | #Pruned Params | Reduction | HBM |
|---|---|---|---|---|---|
| LoRAM-Rand & Stru | 68976648192 | 0.65 | 28099436544 | 2.45× | 52.34 |
| LoRAM-Rand & Stru | 68976648192 | 0.75 | 21488738304 | 3.21× | 40.03 |
| LoRAM-Rand & Stru | 68976648192 | 0.85 | 16272924672 | 4.24× | 30.31 |
| LoRAM-Rand & Stru | 68976648192 | 0.95 | 9662226432 | 7.14× | 18.00 |
| LoRAM-Rand & Stru | 70553706496 | 0.85 | 17849982976 | 3.95× | 33.25 |

Table 6: QLoRAM configures on LLaMA-2-70B and LLaMA-3.1-70B with , demonstrating more aggressive parameter compression.

| Method | #Orig. Params | Pruning Ratio | #Pruned Params | Reduction | HBM |
|---|---|---|---|---|---|
| QLoRAM-Rand & Stru | 68976648192 | 0.65 | 7024859136 | 9.82× | 13.08 |
| QLoRAM-Rand & Stru | 68976648192 | 0.75 | 5372184576 | 12.84× | 10.01 |
| QLoRAM-Rand & Stru | 68976648192 | 0.85 | 4068231168 | 16.95× | 7.58 |
| QLoRAM-Rand & Stru | 68976648192 | 0.95 | 2415556608 | 28.56× | 4.50 |
| QLoRAM-Rand & Stru | 70553706496 | 0.85 | 4462495744 | 15.81× | 8.31 |

## C    VISUALIZATION OF DIMENSION EVOLUTION

To clearly illustrate the evolution of weight matrix dimensions across the multiple stages in the proposed scheme, we take LLM-Pruner (Ma et al., 2023) as an example in (e.g., LoRAM-STRU) in Fig. 9, visualizing the transformation from $\mathbf{W}_0 \Rightarrow \mathbf{W}_0^P$, $\mathbf{W}_\Delta \Rightarrow \mathbf{W}_\Delta^P$, and $\mathbf{W}_\Delta^{P^\star} \Rightarrow \mathbf{W}_\Delta^{R^\star}$ under LoRAM with structured pruning. For LoRAM variants employing non-structured pruning, the parameter dimensionality remains unchanged during training due to the use of a mask matrix. Therefore, these visualizations are omitted.

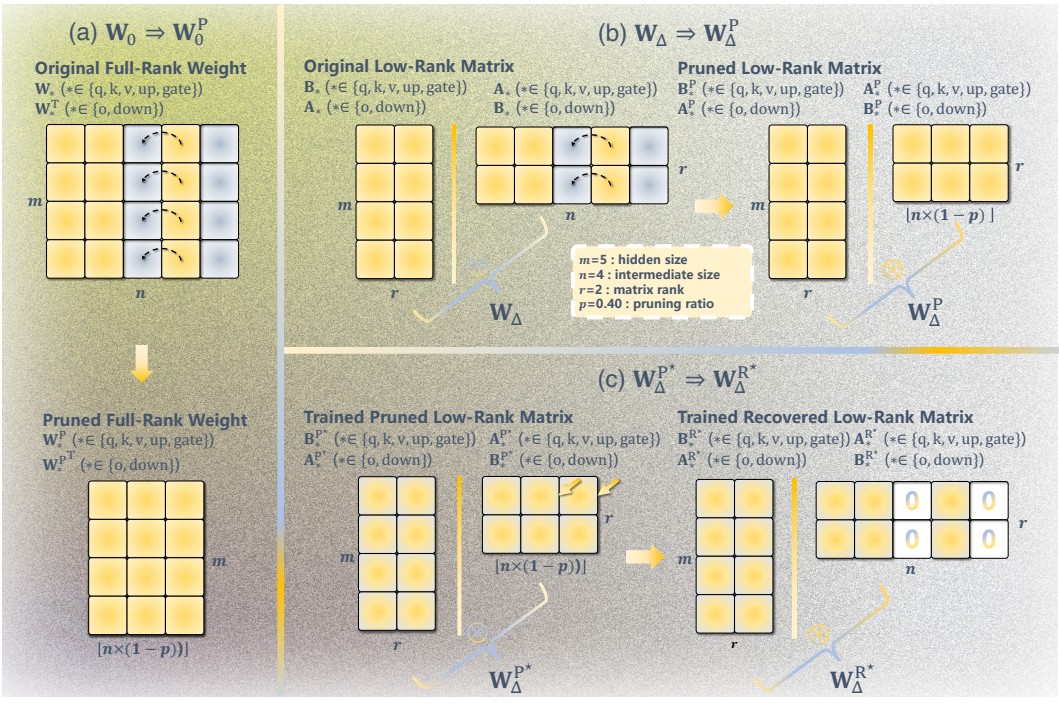

Figure 9: Dimensional evolution of the weight matrices: $\mathbf{W}_0 \Rightarrow \mathbf{W}_0^P$ (a), $\mathbf{W}_\Delta \Rightarrow \mathbf{W}_\Delta^P$ (b), and $\mathbf{W}_\Delta^{P^\star} \Rightarrow \mathbf{W}_\Delta^{R^\star}$ (c) during LoRAM-STRU training. This includes updates for $\mathbf{W}_q$, $\mathbf{W}_k$, $\mathbf{W}_v$, and $\mathbf{W}_o$ in the attention layer, as well as $\mathbf{W}_{up}$, $\mathbf{W}_{gate}$, and $\mathbf{W}_{down}$ in the MLP layer.

# D  VISUALIZATION OF LOW-RANK MATRICES

In this section, we utilize the $L_2$-norm to evaluate variations in low-rank matrices trained with different LoRAM variants. This metric facilitates the visualization of captured features and allows for an analysis of LoRAM's effectiveness. Specifically, we examine the updated low-rank matrices in the self-attention and MLP layers of LLaMA-2-13B and LLaMA-2-70B, trained with LoRAM variants on OpenHermes.

## D.1  HEAD-WISE NORM OF ATTENTION

For the low-rank matrices in the attention layer, denoted as $\mathbf{W}_{\Delta^*}$ where $* \in \{\text{q}, \text{k}, \text{v}, \text{o}\}$, we compute the $L_2$ norms for each attention head. Let $\text{H}^*$ represent the number of heads. The $L_2$ norms for each head h (where $\text{h} = 0, 1, \ldots, \text{H}^* - 1$) are defined as follows:

$$\|\mathbf{W}_{\Delta^*}^{(h)}\|_2 = \begin{cases} \left\|\mathbf{W}_{\Delta^*}[h,:]\right\|_2 & \text{if } * \in \{\text{q}, \text{k}, \text{v}\} \\ \left\|\mathbf{W}_{\Delta^*}[:,h]\right\|_2 & \text{if } * = \text{o} \end{cases}. \tag{10}$$

The results are visualized through heatmaps in Figs. 10 and 12, effectively illustrating the distribution of features captured by different attention heads.

## D.2  LAYER-WISE NORM OF MLP

For the low-rank matrices in the MLP layers, denoted as $\mathbf{W}_{\Delta^*}$ where $\Delta^* \in \{\text{up}, \text{gate}, \text{down}\}$, we denote the number of layers as L. The average $L_2$ norm for a specific layer $l$ (where $l = 0, 1, \ldots, \text{L} - 1$) is computed as follows, excluding elements equal to zero using a mask, ensuring that only active parameters contribute to the average:

$$\|\mathbf{W}_{\Delta^*}^{(l)}\|_2 = \begin{cases} \frac{1}{m} \sum_{i=0}^{m-1} \left\|\mathbf{W}_{\Delta^*}^{(l)}[i,:]\right\|_2 \cdot \mathbb{I}(\mathbf{W}_{\Delta^*}^{(l)}[i,:] \neq 0) & \text{if } \Delta^* \in \{\text{up}, \text{gate}\} \\ \frac{1}{n} \sum_{j=0}^{n-1} \left\|\mathbf{W}_{\Delta^*}^{(l)}[:,j]\right\|_2 \cdot \mathbb{I}(\mathbf{W}_{\Delta^*}^{(l)}[:,j] \neq 0) & \text{if } \Delta^* = \text{down} \end{cases}. \tag{11}$$

Here, $\mathbb{I}(\cdot)$ denotes the indicator function, which returns 1 only when the corresponding element is non-zero, effectively excluding zero elements from the average calculation. The average norms for the MLP layers are visualized in Figs. 11 and 13, clearly depicting the trends in updating amplitudes across the various projections.

## D.3  ATTENTION UPDATE PATTERNS

**Layer Update Patterns in LoRAM and LoRA.**  Figs. 10 and 12 reveal that both LoRA and LoRAM display similar layer update behaviors. In any low-rank matrix $\mathbf{W}_{\Delta^*}$ where $* \in \{\text{q}, \text{k}, \text{v}, \text{o}\}$, deeper colors predominantly concentrate in either shallow or deep layers, while middle layers receive relatively few updates. This suggests that training primarily focuses on optimizing the shallow layers to capture semantic information, with deeper layers refining this knowledge, rendering middle layers somewhat redundant.

**More Uniform Projection Updates in LoRAM.**  Figs. 10 and 12 further indicates that updates in the LoRA-trained low-rank matrices, particularly for $\mathbf{W}_{\Delta^v}$, are relatively uniform, exhibiting substantial deep colors across multiple heads. In contrast, other matrices emphasize specific rows and heads. For instance, in the 70B model's $\mathbf{W}_{\Delta^k}$, only the heads in the uppermost layers experience significant updates, while lower layers show minimal changes. This suggests that the unpruned model retains rich knowledge, requiring only minor adjustments to a few heads in certain layers for task adaptation. Conversely, LoRAM demonstrates a more uniform distribution of deep colors across each low-rank matrix, indicating that the pruned model must effectively utilize every limited neuron to capture knowledge, thereby enhancing downstream performance.

Figure 10: Visualization of low-rank matrices in the attention layers of LLaMA-2-13B.

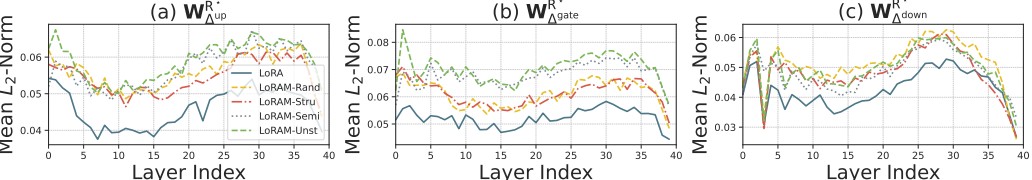

Figure 11: Average $L_2$ norms of low-rank matrices in the MLP layers of LLaMA-2-70B.

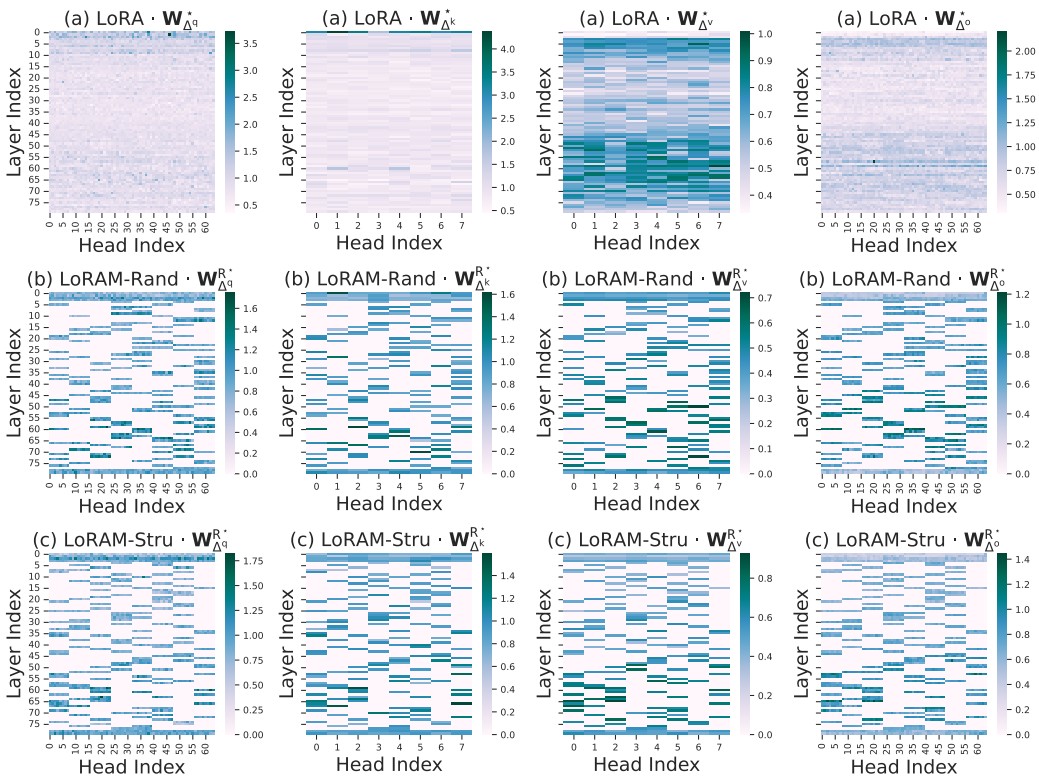

Figure 12: Visualization of low-rank matrices in the attention layers of LLaMA-2-70B.

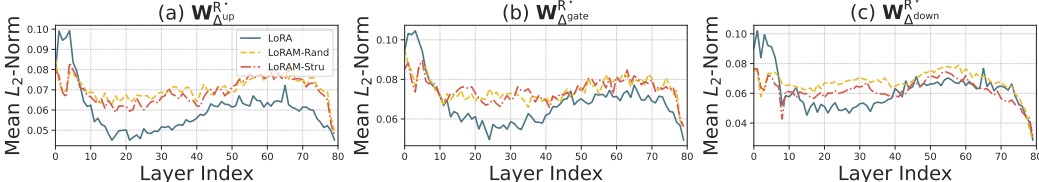

Figure 13: Average $L_2$ norms of low-rank matrices in the MLP layers of LLaMA-2-70B.

## D.4 MLP UPDATE PATTERNS

**LoRAM Exhibits Greater Update Amplitude than LoRA.** For both the 13B and 70B models, LoRAM consistently exhibits a greater update amplitude across each layer compared to LoRA, as shown in Figs. 11 and 13. This increased amplitude indicates that LoRAM is more effective in adjusting the weights in all layers, thus enhancing the adaptability and overall performance.

**Distinct Update Trends in Layer Amplitudes.** The amplitude changes reveal a distinct pattern in Figs. 11 and 13: first decreasing, then increasing, and finally decreasing again. Shallow layers (0-3) and deeper layers (25-35 for the 13B model and 50–75 for the 70B model) undergo intensive updates. This behavior indicates that model prioritizes foundational feature extraction in shallow layers and the refinement of complex representations in deeper layers. Such a strategic update distribution optimizes the learning process, ensuring effective capture of basic and advanced features.

## D.5 ANALYSIS OF UNCHANGED WEIGHTS

Here, we try to analyze the unchanged weights to support the motivation of LoRAM.

**Fine-Grained Visualizations.** As the above visualization, we conducted detailed visualizations comparing the updated magnitudes of pruned and unpruned weights across layers. The results demonstrate that unpruned weights in both attention and MLP layers exhibit consistently smaller updates during fine-tuning as shown in Fig. 12, indicating their critical role in preserving the model's capacity for inference.

**Theoretical Perspective.** The phenomenon can be explained by the gradient-based importance of these weights, which prioritize parameters with minimal updates but high sensitivity during recovery. These weights stabilize inference outputs, making them indispensable despite their limited fine-tuning updates.

**Quantitative Evidence** Our analysis reveals a strong correlation between weight update magnitudes and downstream performance. Pruning weights with smaller updates significantly degrades performance, highlighting their importance for inference and validating our intuition.

**Impact on Large Models** The selective pruning strategy shows notable benefits in larger models such as LLaMA-2-70B, where it outperforms random pruning by a substantial margin. Retaining critical parameters ensures effective task adaptation and generalization across diverse domains.

# E    PERFORMANCE OF SUB-TASKS IN CSR

We report the performance of six sub-tasks in CSR, with Figs. 14 and 15 showcasing the results for LORAM-trained LLaMA-2-13B and LLaMA-2-70B, respectively. Our findings indicate that various LORAM variants outperform core competitive benchmarks: for the 13B model, LORAM surpasses both the untrained 13B and the LoRA-trained 7B, while for the 70B model, it exceeds the untrained 70B and the LoRA-trained 13B. This demonstrates that LORAM consistently achieves performance gains across models of different scales while effectively reducing memory usage. Furthermore, selective weight contributions in the 70B model significantly enhance performance, as evidenced by LORAM-STRU's marked improvement, particularly in the challenging Arc Challenge multi-choice question-answering task. This suggests that LORAM-STRU effectively identifies and leverages weight differences, focusing on the most trainable weights compared to LORAM-RAND.

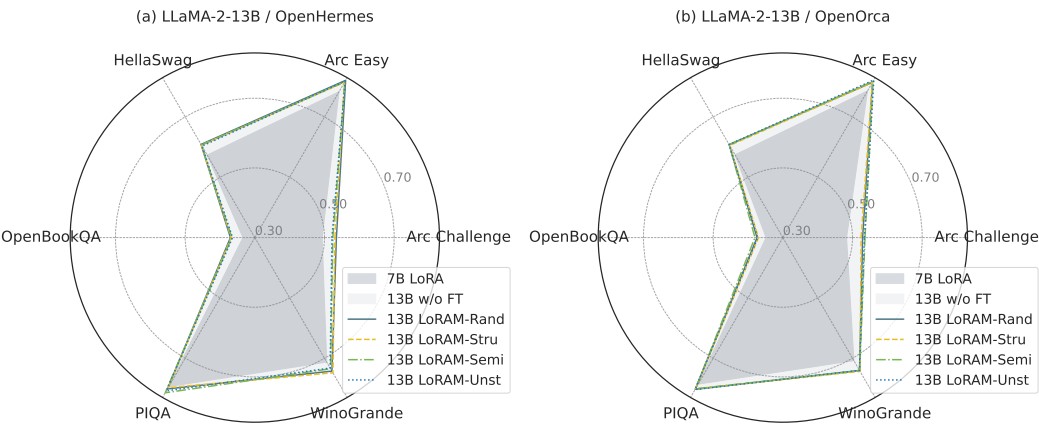

Figure 14: Performance of six CSR sub-tasks on the trained LLaMA-2-13B using LORAM.

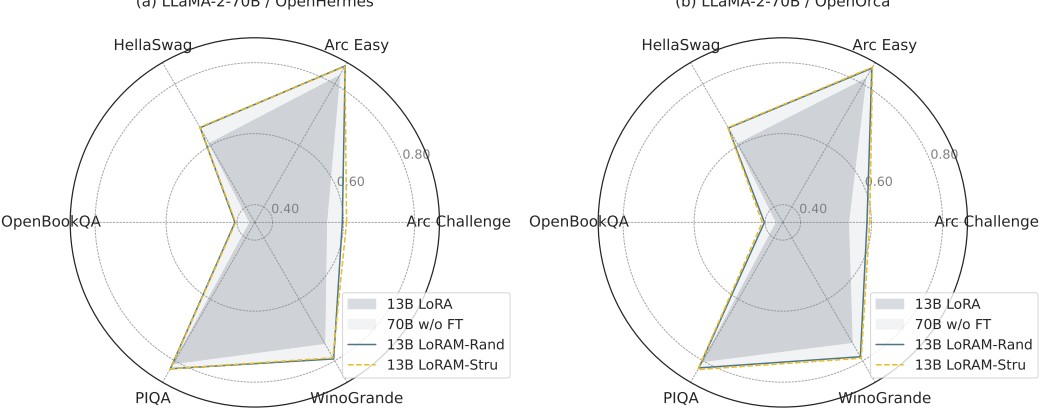

Figure 15: Performance of six CSR sub-tasks on the trained LLaMA-2-70B using LORAM.

# F  ALGORITHM OF LORAM

Here, we present the complete algorithm of LoRAM in Algorithm 1.

---

**Algorithm 1** LoRAM (Memory-Efficient LoRA Training)

---

**Require:** original full-rank pre-trained weight $\mathbf{W}_0$, alignment corpus $\mathcal{D}_{\mathtt{A}}$, and flags $\mathcal{F}^{\mathtt{P}}, \mathcal{F}^{\mathtt{A}}, \mathcal{F}^{\mathtt{Q}}, \mathcal{F}^{\mathtt{R}}$.

1: **Offline $\mathbf{W}_0^*$ Process Stage:**
2: **if** $\mathcal{F}^{\mathtt{P}}$ **then**
3:      $\mathbf{W}_0^{\mathtt{P}} = \mathtt{P}(\mathbf{W}_0) = \mathbf{W}_0 \circ \mathbf{M}^{\mathtt{P}}$            ▷ Pruned Full-Rank Weight Generation.
4:      **if** $\mathcal{F}^{\mathtt{A}}$ **then**
5:          $\mathbf{W}_0^{\mathtt{P},\mathtt{A}} \leftarrow \arg\min \mathcal{L}_{\mathtt{A}}(\mathcal{D}_{\mathtt{A}}; \mathbf{W}_0^{\mathtt{P}})$       ▷ Pruned Full-Rank Weight Alignment.
6:          **if** $\mathcal{F}^{\mathtt{Q}}$ **then**
7:              $\mathbf{W}_0^{\mathtt{P},\mathtt{A},\mathtt{Q}} = \mathtt{Q}(\mathbf{W}_0^{\mathtt{P},\mathtt{A}})$       ▷ Pruned Full-Rank Weight Quantization.
8:          **end if**
9:      **else if** $\mathcal{F}^{\mathtt{Q}}$ **then**
10:          $\mathbf{W}_0^{\mathtt{P},\mathtt{Q}} = \mathtt{Q}(\mathbf{W}_0^{\mathtt{P}})$
11:      **end if**
12: **else if** $\mathcal{F}^{\mathtt{Q}}$ **then**
13:      $\mathbf{W}_0^{\mathtt{Q}} = \mathtt{Q}(\mathbf{W}_0)$              ▷ Standard Quantization for LoRA
14: **end if**
15: Record the processing result of $\mathbf{W}_0$ as $\mathbf{W}_0^*$, $* \in \{\mathtt{NULL}, \mathtt{P}, \mathtt{Q}, (\mathtt{P}, \mathtt{Q}), (\mathtt{P}, \mathtt{A}), (\mathtt{P}, \mathtt{A}, \mathtt{Q})\}$.
16:
17: **Online $\mathbf{W}_{\triangle}^*$ Training Stage:**
18: **if** $\mathcal{F}^{\mathtt{P}}$ **then**                ▷ Pruned Low-Rank Matrix Generation.
19:      $\mathbf{W}_{\triangle}^{\mathtt{P}} = \mathbf{B}^{\mathtt{P}}\mathbf{A}^{\mathtt{P}} = \mathtt{P}(\mathbf{W}_{\triangle}) = \mathbf{W}_{\triangle} \circ \mathbf{M}^{\mathtt{P}} = \mathbf{B}\mathbf{A} \circ \mathbf{M}^{\mathtt{P}}$
20:      **while** TRAINING **do**            ▷ Pruned Low-Rank Matrix Training.
21:          Update low-rank matrix via objective $\mathcal{L}_{\mathrm{SFT}}$ with the forward pass $\mathbf{h} = \mathbf{x}\mathbf{W}_0^* + \mathbf{x}\mathbf{W}_{\triangle}^{\mathtt{P}}$.
22:          Return trained low-rank matrix $\mathbf{W}_{\triangle}^{\mathtt{P}^\star} = \mathbf{B}^{\mathtt{P}^\star}\mathbf{A}^{\mathtt{P}^\star}$.
23:      **end while**
24:      **if** $\mathcal{F}^{\mathtt{R}}$ **then**         ▷ Recovered Low-Rank Matrix Generation.
25:          $\mathbf{W}_{\triangle}^{\mathtt{R}^\star} = \mathbf{B}^{\mathtt{R}^\star}\mathbf{A}^{\mathtt{R}^\star} = \mathtt{R}(\mathbf{W}_{\triangle}^{\mathtt{P}^\star}) = \mathbf{W}_{\triangle}^{\mathtt{P}^\star} \circ (1 - \mathbf{M}^{\mathtt{P}})$     ▷ Structured LoRAM
26:      **else**
27:          $\mathbf{W}_{\triangle}^{\mathtt{R}^\star} = \mathbf{B}^{\mathtt{R}^\star}\mathbf{A}^{\mathtt{R}^\star} = \mathbf{B}^{\mathtt{P}^\star}\mathbf{A}^{\mathtt{P}^\star}$       ▷ Non-structured LoRAM
28:      **end if**
29: **else**
30:      **while** TRAINING **do**             ▷ Standard LoRA Training.
31:          Update low-rank matrix via objective $\mathcal{L}_{\mathrm{SFT}}$ with the forward pass $\mathbf{h} = \mathbf{x}\mathbf{W}_0^* + \mathbf{x}\mathbf{W}_{\triangle}$.
32:          Return trained low-rank matrix $\mathbf{W}_{\triangle}^\star = \mathbf{B}^\star\mathbf{A}^\star$.
33:      **end while**
34: **end if**
35: Record the trained low-rank matrix as $\mathbf{W}_{\triangle}^*$, $* \in \{\mathtt{R}^\star, \star\}$.
36:
37: **Online $\mathbf{W}_0, \mathbf{W}_{\triangle}^*$ Inference Stage:**
38: **while** INFERENCE with $*$ is $\mathtt{R}^\star$ **do**         ▷ Recovered Low-Rank Matrix Inference.
39:      Perform inference with the forward pass $\mathbf{h} = \mathbf{x}(\mathbf{W}_0 + \mathbf{W}_{\triangle}^{\mathtt{R}^\star}) = \mathbf{x}(\mathbf{W}_0 + \mathbf{B}^{\mathtt{R}^\star}\mathbf{A}^{\mathtt{R}^\star})$.
40: **end while**
41: **while** INFERENCE with $*$ is $\star$ **do**             ▷ Standard LoRA Inference.
42:      Perform Inference with the forward pass $\mathbf{h} = \mathbf{x}(\mathbf{W}_0 + \mathbf{W}_{\triangle}^\star) = \mathbf{x}(\mathbf{W}_0 + \mathbf{B}^\star\mathbf{A}^\star)$.
43: **end while**

---

## G   TUNING OF LEARNING RATE

We provide additional details on the learning rate tuning process for full LoRA applied to LLaMA-2-7B and LLaMA-2-13B models, trained on the OpenHermes dataset. These experiments in Fig. 16 demonstrate that a learning rate of 1e-3 consistently achieves the best perplexity across both in-domain and out-of-domain datasets, further validating the reliability of our comparison.

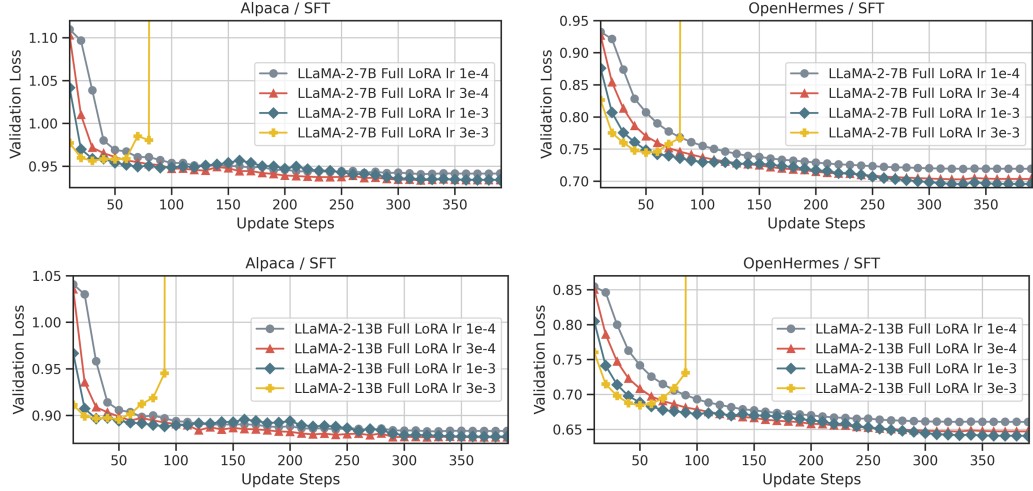

Figure 16: Learning rate tuning for LLaMA-2-7B and LLaMA-2-13B on OpenHermes using LoRA.

## H   PERFORMANCE OF DOMAIN-SPECIFIC TASK

To assess the effectiveness of LoRAM in domain-specific tasks, we conducted experiments on GSM8K (using the training set for tuning and the test set for evaluation), a mathematical reasoning benchmark known for its sensitivity to sparsification. Specifically, we trained LLaMA-3.1-70B using QLoRAM under various configurations.

The results, summarized in Table 7, highlight that LoRAM achieves excellent performance in this domain-specific setting. Notably, LoRAM-based models maintain high accuracy with substantial parameter reduction ratios, showcasing their robustness and efficiency in domain-specific tasks. These findings emphasize LoRAM's broad applicability beyond general-purpose instruction fine-tuning.

Table 7: Evaluation of LoRAM on the GSM8K dataset for domain-specific fine-tuning. Results show accuracy (%) and parameter reduction ratios for different configurations.

| LLaMA-3.1 | GSM8K | Parameter Reduction Ratio |
|---|---|---|
| 8B w/o Fine-Tuning | 55.27 | 8.79× |
| 8B LoRA (OpenHermes 400) | 55.80 | 8.79× |
| 70B w/o Fine-Tuning | 75.28 | 1.00× |
| 70B QLoRAM-Stru 400 (OpenHermes 400) | 80.36 | **15.81×** |
| 70B QLoRAM-Stru 400 (GSM8K 100) | 77.18 | **15.81×** |
| 70B QLoRAM-Stru 400 (GSM8K 200) | 79.15 | **15.81×** |
| 70B LoRA (OpenHermes 400) | **80.74** | 1.00× |

# I    ANALYSIS OF LoRAM COST

Identifying the costs of LoRAM is indeed important, which is why we report both the number of training tokens used during the alignment phase and the parameter reduction ratios in the low-rank training phase. Below, we clarify the two stages of LoRAM:

**Offline Knowledge Alignment Phase.**    The offline phase is task-agnostic and can be conducted by the model publisher prior to deployment, making its cost negligible for end users. To quantify the offline cost, we measured the number of training tokens (as in Xia et al. (2024)) rather than end-to-end latency, which can vary based on hardware configurations. As shown in Figure 5, LoRAM achieves significant performance gains using only 13 million tokens, demonstrating the efficiency of the alignment phase.

**Online Low-Rank Matrix Training Phase.**    For the online phase, the memory and latency costs are primarily determined by the size of the base model parameters, which dominate resource consumption during training. To avoid redundancy in reporting, we focused on parameter reduction ratios instead of absolute time or memory usage.

**Comparative Metrics for Online Training.**    Here, we provide additional metrics, including memory and latency comparisons for the online training phase. We conducted experiments using a workload of 1024 samples (batch size 128, micro-batch size 4, sequence length 512) randomly selected from OpenHermes. The results in Table 8 demonstrate that LoRAM with a structured pruning ratio of $2.17\times$ (13B $\rightarrow$ 6B) achieves comparable peak memory, latency, and throughput to 7B LoRA, with only minor trade-offs. These differences arise due to the larger layer count in 13B LoRAM, introducing more non-GEMM operations, slightly affecting latency and throughput.

These results underscore the advantages of LoRAM's design in achieving substantial resource efficiency without significant trade-offs in memory or latency.

Table 8: Comparison of peak memory (MiB), latency (s), and throughput (samples/s) during the online training phase for LoRAM and LoRA models. Results are based on a workload of 1024 samples (batch size 128, micro-batch size 4, sequence length 512).

| LLaMA-2 | #Model Params | Reduction Ratio | Memory | Latency | Throughput |
|---|---|---|---|---|---|
| 7B LoRA | 6.73B | 1.93× | 30,517 | **134.27** | **7.626** |
| 13B LoRA | 13.02B | 1.00× | 51,661 | 206.07 | 4.969 |
| 13B LoRAM-Stru | **6.01B** | **2.17×** | **29,799** | 147.86 | 6.925 |

## J    ANALYSIS OF CHANGES IN PERFORMANCE TRENDS

We analyze performance at two stages: after fine-tuning but before recovery, and after both fine-tuning and recovery.

**After Fine-Tuning but Before Recovery.**    At this stage, the results of LoRAM align with prior work (e.g., SparseGPT, Wanda, and LLM-Pruner). Unstructured and semi-structured pruning consistently outperform structured pruning (see Fig. 6, solid lines). This trend holds true across both aligned and unaligned settings, with the performance order as follows: LoRAM-SEMI < LoRAM-UNST < LoRAM-STRU < LoRAM-RAND The slight advantage of LoRAM-SEMI over LoRAM-UNST can be attributed to its smaller pruning ratio, which retains more parameters and mitigates performance degradation.

**After Fine-Tuning and Recovery.**    Post-recovery results show that structured pruning outperforms unstructured pruning. This can be explained by two factors:

- **Preserved Structure for Recovery:** Structured pruning maintains the organization of the pruned weights into coherent structures (e.g., rows and columns in MLP layers, attention heads in attention layers), ensuring that activations after recovery are aligned with those of the original model. This alignment improves the recovery process.

- **Pruned Weight Quality:** The quality of pruned weights influences the recovery effectiveness. Structured pruning tends to remove less critical weights, leaving more recoverable parameters. In contrast, unstructured pruning can remove weights that are more difficult to recover, which negatively impacts performance post-recovery.

These results highlight the interplay between pruning and recovery, suggesting that structured pruning, despite initial performance disadvantages, facilitates more effective recovery.

