# OpenReview forum: "Train Small, Infer Large: Memory-Efficient LoRA Training for Large Language Models"
_ICLR.cc/2025/Conference — ICLR 2025 Poster_

### Official Review · Reviewer_BSMa · 2024-10-22

**Soundness:** 3
**Presentation:** 4
**Contribution:** 3
**Rating:** 8
**Confidence:** 5

**Summary:**

This work proposes a memory-efficient training scheme for LLMs, named LoRAM. Based on an intuition that some parameters remain unchanged during finetuning, but are important during inference, the authors train a pruned model by updating the pruned low-rank matrices and then uses recovered matrices for inference. In this way, the parameters need to be finetuned can be largely reduced while achieving similar accuracy compared with previous LoRA methods.

**Strengths:**

-Memory efficient training is an important topic in LLM-related fields.

-This paper proposes an easy but effective method to reduce the number of parameters which need to be finetuned.

-The potential this sparsity-based method shows when combined with QLoRA seems a promising way for this field.

-The paper is easy to follow, experiments presented are good.

**Weaknesses:**

-My major concern is that why cannot train small and infer small? Training small is convincing enough, but it sounds more reasonable to me that we train on the pruned model, and then also infer on that model. I think it is indispensable for this paper to show the certain benefits of inferring large over inferring small. Detailedly, experimental results (e.g. model accuracy or inference latency) are needed for the comparison of inferring small and inferring large, so that the inference design can be convincing.

-This idea is based on the intuition that some parameters remain unchanged during finetuning, but are important during inference, although the proposed method seems to work, there is no evidence to show the phenomenon and no theoretical analysis to show the reason of the phenomenon. What is the property of those unchanged weights? This paper will be much better with more effort on this part.

**Questions:**

-What is the training throughput of your method? Updating pruned parameters seems time-consuming generally.

---

> ### Author Response · Authors · 2024-11-21
> **Thanks & Initial Response to Reviewer BSMa (W1,W2)**
>
> Identifying the trade-offs of "train small, infer small" is indeed crucial, and its performance implications are already reflected in the **Ablation Study** of our paper.
>
> **1. Performance Impact of Recovery:**
>    The "train small, infer small" approach essentially corresponds to a variant of LoRAM without recovery (w/o Recovery). However, this approach is not adopted due to its **significantly inferior performance**.
>
> **2. Validation on Alpaca PPL:**
>    In Section 4.5, we evaluated the impact of recovery on perplexity (PPL). As shown in Figure 6, recovery consistently outperforms the unrecovered approach by using LLaMA-2-13B with LoRAM-Stru on the Alpaca test set.
>    - **With alignment (w/ Alignment):** Recovery achieves a PPL nearly **1.4× lower** than w/o Recovery.
>    - **Without alignment (w/o Alignment):** This gap widens to approximately **1.85×**.
>
> Such a significant PPL degradation would inevitably lead to noticeable performance drops in downstream tasks, making recovery an indispensable part of our method's design.
>
> ---
>
> Thank you for recognizing the significance and effectiveness of our work. We acknowledge the need for more explicit evidence and theoretical insights regarding the properties of unchanged weights. Below, we provide additional analysis to address your concerns:
>
> **1. Fine-Grained Visualizations:**
>    We conducted detailed visualizations comparing the updated magnitudes of pruned and unpruned weights across layers. The results demonstrate that unpruned weights in both attention and MLP layers exhibit consistently smaller updates during fine-tuning, indicating their critical role in preserving the model's capacity for inference.
>
> **2. Theoretical Perspective:**
>    The phenomenon can be explained by the gradient-based importance of these weights, which prioritize parameters with minimal updates but high sensitivity during recovery. These weights stabilize inference outputs, making them indispensable despite their limited fine-tuning updates.
>
> **3. Quantitative Evidence:**
>    Our analysis reveals a strong correlation between weight update magnitudes and downstream performance. Pruning weights with smaller updates significantly degrades performance, highlighting their importance for inference and validating our intuition.
>
> **4. Impact on Large Models:**
>    The selective pruning strategy shows notable benefits in larger models such as LLaMA-2-70B, where it outperforms random pruning by a substantial margin. Retaining critical parameters ensures effective task adaptation and generalization across diverse domains.
>
> These findings enhance the theoretical and empirical basis of our method, and we plan to highlight them in the revisions.

---

> > ### Comment · Reviewer_BSMa · 2024-11-22
> > **Response to the authors**
> >
> > Thanks for your responding.
> >
> > The title of your paper mentions "infer large," but the concept of "large" and "small" is hardly discussed further in the main text. This might cause readers to overlook some key aspects of your design. I suggest using the terms "infer large" and "infer small" more frequently throughout the paper. Moreover, you should clearly point out that the "unrecovered" model refers to the small model and the "recovered" model refers to the large model. You could even include a diagram from a high-level perspective to illustrate the advantages of the large model over the small model, making it easier for readers to follow your inference design.
> >
> > Additionally, if your findings on "unchanged weight" are well supported by experimental results, your paper would present a complete piece of work. I further recommend referencing the writing style of Fig.3 in PB-LLM [1] paper during your revision. Specifically, you can use figures and data at the beginning to clearly highlight your relevant discoveries. This would make your insights much stronger.
> >
> > If you include a detailed analysis of "unchanged weights" in your revision, I will consider raising my score.
> >
> > [1] https://arxiv.org/pdf/2310.00034

---

> ### Author Response · Authors · 2024-11-21
> **Thanks & Initial Response to Reviewer BSMa (Q1)**
>
> **[Q1-What is the training throughput of your method? Updating pruned parameters seems time-consuming generally.]**
>
>
> Training cost is indeed a critical consideration, which is why we use **the number of training tokens** as the metric to measure it. This metric is hardware-agnostic, unlike end-to-end training time. Below, we address this concern by clarifying LoRAM's phases, validating its task-agnostic alignment, and discussing the impact of corpus size.
>
> **1. Offline Phase Clarification:**
>    LoRAM comprises an **offline alignment phase** and an **online fine-tuning phase**. The offline phase updates pruned parameters to align the pruned model with the original one, as detailed in Section 3.2. This step is performed once on a **small, task-agnostic corpus** by the model publisher, making it resource-efficient. End users only need to fine-tune low-rank matrices on the aligned pruned model for their downstream tasks, avoiding the computational cost of alignment.
>
> **2. Task-agnostic Validation:**
>    The task-agnostic nature of LoRAM's alignment was validated across various LLaMA-2 model scales and fine-tuning corpora (Sections 4.2 and 4.3). Since alignment is performed once offline, it **does not burden user resources**. Training throughput was measured in terms of **tokens processed** (as in ShearLLaMA [1]). For instance, aligning the 70B QLoRAM-Stru model required a **13M-token corpus** (~200 steps), achieving strong downstream task performance with minimal cost.
>
> **3. Corpus Size Impact:**
>    As explored in Section 4.4, the impact of corpus size on performance demonstrates LoRAM's efficiency. For a 70B model with a **7.07× parameter reduction**, alignment on 13M tokens already delivered results competitive with LoRA-trained models. Doubling the corpus size to 26M tokens led to further performance gains, surpassing LoRA-trained 70B in certain domains like MathQA.
>
> |                     | MathQA    | GSM8K     | CSR       | HumanEval (Pass@10) |
> | ------------------- | --------- | --------- | --------- | ------------------- |
> | 8B w/o FT           | 42.21     | 55.27     | 64.77     | 61.59               |
> | 8B LoRA             | 41.14     | 55.80     | 65.32     | 71.34               |
> | 70B w/o FT          | 54.65     | 75.28     | 70.52     | 84.76               |
> | 70B LoRA            | 51.66     | **80.74** | 70.63     | 84.15               |
> | 70B QLoRAM-Stru 200 | **54.67** | 79.40     | 70.55     | 83.54               |
> | 70B QLoRAM-Stru 400 | 54.64     | 80.36     | **70.71** | **85.37**           |
>
> These results highlight LoRAM's ability to balance computational efficiency with competitive performance, making it a practical and scalable solution for resource-constrained users.
>
>
>
> Additionally, we recognize that other performance metrics beyond parameter reduction ratio may also be of interest. Below, we provide a detailed comparison using LLaMA-2-13B as an example.
>
> The table below, based on a workload of 1024 samples (batch size 128, micro-batch size 4, sequence length 512) from OpenHermes, demonstrates that **LoRAM with a structured pruning ratio of 2.17× (13B → 6B)** achieves comparable **peak memory**, **latency**, and **throughput** to the **7B LoRA model**, with only minor trade-offs. These slight differences arise from the larger layer count in 13B LoRAM, which introduces slightly more non-GEMM operations.
>
> This comparison underscores LoRAM’s capability to **significantly reduce resource demands** while maintaining competitive efficiency, making it a practical choice for resource-constrained environments.
>
> |                        | Parameters Size of Base Model | Parameter Reduction Ratio | Peak Memory (MiB) | Latency (s) | Throughput (samples/s) |
> | ---------------------- | ----------------------------- | ------------------------- | ----------------- | ----------- | ---------------------- |
> | LLaMA-2-7B LoRA        | 6.74B                         | 1.93×                     | 30,517            | **134.27**  | **7.626**              |
> | LLaMA-2-13B LoRA       | 13.02B                        | 1.00×                     | 51,661            | 206.07      | 4.969                  |
> | LLaMA-2-13B LoRAM-Stru | **6.01B**                     | **2.17×**                 | **29,799**        | 147.86      | 6.925                  |

---

> ### Author Response · Authors · 2024-11-22
> **Follow up Response to Reviewer BSMa**
>
> Thank you for your thoughtful suggestions. Below, we address the key concerns regarding the clarity of "large" and "small" model terminology and the analysis of unchanged weights:
>
> **1. Clarification of "Infer Large" and "Infer Small"**
> We agree that clearer terminology will help readers better understand our design. To address this, we have updated the abstract to explicitly define the correspondence between the "large" model and the recovered full model, and the "small" model and the pruned model. In the main text, we will incorporate more consistent usage of these terms and include a high-level diagram contrasting the advantages of the large model over the small one in appendix, highlighting the explanation of our inference design.
>
> **2. Analysis of Unchanged Weights**
> PB-LLM serves as an excellent reference; however, due to constraints in the available revision time and manuscript space, a complete restructuring of the paper is challenging. Nonetheless, we have incorporated our findings on unchanged weights in Appendix C.5 to support the motivation for LoRAM and have referenced these findings in the main text for better visibility.
>
> Thank you for your constructive suggestions. These revisions are designed to improve clarity and highlight the significance of our contributions. We hope these updates address your concerns effectively and look forward to your further feedback.

---

> > ### Comment · Reviewer_BSMa · 2024-11-22
> > **Follow up Response to the authors**
> >
> > I decide to raise my score.

---

> > > ### Author Response · Authors · 2024-11-22
> > > **Follow up Response to Reviewer BSMa**
> > >
> > > Thanks for your rasing score, it’s a huge recognition of our work!

---

### Official Review · Reviewer_H1mX · 2024-11-01

**Soundness:** 2
**Presentation:** 3
**Contribution:** 3
**Rating:** 6
**Confidence:** 3

**Summary:**

The paper proposes LoRAM, a memory-efficient training scheme that combines pruning, LoRA-based training on the pruned model and subsequent integration of the found solution back in the original model. This procedure yields significant memory reduction compared to LoRA-based fine-tuning, which in turn was already an improvement over the full-model fine-tuning.

Authors further observed that under aggressive pruning rates their method did not work well and identified that the main culprit is inconsistency between the pruned model used for training and the original model used for inference. To combat this, they train the pruned model on a relatively small corpus to achieve alignment, which is a one-shot offline process.

The effectiveness of LoRAM is evaluated across several model sizes, pruning algorithms, and tasks in different domains.

**Strengths:**

* The paper is fairly well structured and well written. Visualizations are quite helpful for quickly grasping the method.
* LoRAM enables training a 70B model on a GPU with only 20GB, which is an impressive result.
* The proposed method is complementary with other memory reduction techniques such as quantization and pruning strategies.

**Weaknesses:**

* Some comparisons in Tables 1-3 seem unfair: It would be nice to include a baseline that uses the same total training time/compute as LoRAM (including alignment & LoRA-based fine-tuning of the pruned model), but for full fine-tuning of the original model.
* Aside from the trivial “w/o FT” baseline, it seems that LoRA is the only method LoRAM is compared against. It would be nice to include more baselines from the literature. For instance, would be interesting to compare LoRAM to the respective pruning method used to see the effect of subsequent LoRA training.
* The paper did not report the training speed LoRAM compared to LoRA and full-model fine-tuning. Having a comparison of wall-clock time and/or throughput (e.g. tokens/second) for LoRAM vs LoRA and full fine-tuning across a few model sizes would give a more complete picture of the trade-offs.
* It would greatly help to increase the reproducibility of this work by releasing the code.

**Questions:**

* In Figure 3 & 4, any reason for instability for QLoRAM-Rand & QLoRAM-Stru for 70B model when evaluated on Alpaca?
* Seems there is a typo in Table 3: 0.3415 should be 34.15

---

> ### Author Response · Authors · 2024-11-21
> **Thanks & Initial Response to Reviewer H1mX (W1)**
>
> **[W1-Some comparisons in Tables 1-3 seem unfair: It would be nice to include a baseline that uses the same total training time/compute as LoRAM (including alignment & LoRA-based fine-tuning of the pruned model), but for full fine-tuning of the original model.]**
>
> We appreciate the reviewer’s suggestion and would like to clarify the rationale behind the comparisons in Tables 1–3, as well as the challenges in introducing the proposed baseline.
>
>  **1. Resource-Constrained User Perspective**:
>    The comparisons in Tables 1–3 aim to reflect realistic scenarios where users operate under limited resources. Specifically, we analyze whether training a smaller model with LoRA or adopting LoRAM to train a larger model is the more cost-effective choice. The results consistently demonstrate that **LoRAM offers better performance-efficiency trade-offs under constrained budgets**. Even without the proposed baseline, LoRAM outperforms **w/o FT** significantly and, in some cases, even surpasses full LoRA fine-tuning of equivalent-scale models (e.g., LLaMA-3.1-70B), as shown in the below table.
>
> **2. Challenges with the Proposed Baseline**:
>    Introducing the baseline suggested by the reviewer poses two key challenges:
>    - **Resource Limitations**: Full fine-tuning of the original large-scale model (e.g., LLaMA-3.1-70B) is computationally prohibitive. For example, fine-tuning a 70B model requires **1178GB of memory**, necessitating at least **15 A100 GPUs**, which exceeds our experimental capacity. Furthermore, the extremely high cost of full fine-tuning falls outside the resource-constrained scenarios targeted by our work.
>    - **Clarifying the Alignment Cost**: The alignment phase in LoRAM is **task-agnostic and can be performed once offline** by the model provider. This cost is not borne by the end user and is conceptually distinct from the user's fine-tuning objective. As such, comparing against full fine-tuning of the original model, which involves entirely different cost dynamics, would not provide a meaningful evaluation of LoRAM's design.
>
> By leveraging LoRA on structured pruned models, LoRAM serves as a **practical and cost-efficient solution**, particularly for users with limited computational resources. While the suggested baseline is conceptually interesting, it diverges from the resource-constrained use cases central to our study.
>
> | Model               | GSM8K        | Parameter Reduction Ratio |
> | ------------------- | ------------ | ------------------------- |
> | 8B w/o FT           | 55.27        | **8.79×**                 |
> | 8B LoRA             | 55.80        | **8.79×**                 |
> | 70B w/o FT          | 75.28        | 1.00×                     |
> | 70B QLoRAM-Stru 400 | 80.36 |7.07×              |
> | 70B LoRA            | **80.74**    | 1.00×                     |

---

> ### Author Response · Authors · 2024-11-21
> **Thanks & Initial Response to Reviewer H1mX (W2,W3,W4)**
>
> **[W2-Aside from the trivial “w/o FT” baseline, it seems that LoRA is the only method LoRAM is compared against. It would be nice to include more baselines from the literature. For instance, would be interesting to compare LoRAM to the respective pruning method used to see the effect of subsequent LoRA training.]**
>
> We agree that comparing LoRAM with baseline methods that combine different pruning strategies followed by LoRA training (naive pruning) is highly important. This comparison is included in our ablation study, as these baselines essentially represent LoRAM variants without the recovery phase (w/o Recovery).
>
> Our results show that across different pruning strategies, these variants exhibit significantly worse perplexity (PPL) compared to the complete LoRAM pipeline. For instance, LoRAM-Stru (w/o Alignment) shows **a gap of nearly 1.8 in PPL**. As widely acknowledged, such a degree of PPL degradation would substantially impact downstream task performance.
>
> ---
>
> **[W3-The paper did not report the training speed LoRAM compared to LoRA and full-model fine-tuning. Having a comparison of wall-clock time and/or throughput (e.g. tokens/second) for LoRAM vs LoRA and full fine-tuning across a few model sizes would give a more complete picture of the trade-offs.]**
>
> We agree that comparing the training speed of LoRAM with LoRA and full fine-tuning provides a more complete picture of trade-offs. While we initially focused on **parameter reduction ratios** to highlight memory and latency advantages, we conducted additional experiments to address this concern directly.
>
> The results in the table below, based on a workload of 1024 samples (batch size 128, micro-batch size 4, sequence length 512) from OpenHermes, show that **LoRAM with a structured pruning ratio of 2.17× (13B → 6B)** achieves comparable peak memory, latency, and throughput to **7B LoRA**, with only minor trade-offs. These differences are due to the larger layer count in 13B LoRAM, introducing slightly more non-GEMM operations.
>
> This comparison highlights LoRAM’s ability to significantly reduce resource requirements while maintaining competitive efficiency.
>
> |                        | Parameters Size of Base Model | Parameter Reduction Ratio | Peak Memory (MiB) | Latency (s) | Throughput (sample/s) |
> | ---------------------- | ----------------------------- | ------------------------- | ----------------- | ----------- | --------------------- |
> | LLaMA-2-7B LoRA        | 6738415651                    | 1.93×                     | 30517             | **134.27**  | **7.626**             |
> | LLaMA-2-13B LoRA       | 13015864320                   | 1.00×                     | 51661             | 206.07      | 4.969                 |
> | LLaMA-2-13B LoRAM-Stru | **6005662720**                | **2.17×**                 | **29799**         | 147.86      | 6.925                 |
>
> ---
>
>
> **[W4-It would greatly help to increase the reproducibility of this work by releasing the code.]**
>
> Thank you for acknowledging the significance of our work. LoRAM offers a simple yet highly effective solution to **significantly reduce the training cost for resource-constrained users** aiming for customized fine-tuning. To further enhance its accessibility and reproducibility, we will release an initial version of our codebase, enabling the community to leverage and build upon our approach.
>
> ----
>
> ## **`Update 11.24 (Anonymous Version of LoRAM Execution Code):`**
>
> We are delighted to share that an anonymous version of the LoRAM execution code has been uploaded to the supplementary materials. We sincerely hope you find the features of LoRAM both fascinating and highly practical, as it significantly reduces training memory usage while achieving impressive inference performance gains.
>
> This version is relatively simple due to time constraints, but we are committed to addressing any issues you might encounter and plan to release an even more user-friendly implementation in the future. **If you know users with resource constraints who could benefit from task-specific customization, LoRAM is definitely worth recommending!** Your feedback and suggestions are always welcome, and we deeply appreciate your engagement with our work!

---

> ### Author Response · Authors · 2024-11-21
> **Thanks & Initial Response to Reviewer H1mX (Q1, Q2)**
>
> **[Q1-In Figure 3 & 4, any reason for instability for QLoRAM-Rand & QLoRAM-Stru for 70B model when evaluated on Alpaca?]**
>
> The observed instability for QLoRAM-Rand and QLoRAM-Stru on the 70B model aligns with our conclusion in Section 4.2: *Non-Structured LoRAM Excels in In-Domain.* Structured pruning inherently struggles to preserve the information capture capabilities of the original model, leading to greater variability in out-of-domain evaluation performance.
>
> ----
>
> **[Q2-Seems there is a typo in Table 3: 0.3415 should be 34.15]**
>
> Thank you for your meticulous review and positive feedback on the overall writing and presentation of our work. We have corrected the typo in Table 3 (0.3415 → 34.15) and highlighted the change in blue in the revised version uploaded for your reference.

---

> ### Author Response · Authors · 2024-11-25
> **Follow-Up: Seeking Further Feedback (Discussion Deadline Approaching in 2 Days)**
>
> Dear Reviewer H1mX,
>
> We hope this message finds you well. Following up on our recent exchange regarding this paper, we wanted to kindly check if there are any further concerns or feedback from you. With the discussion deadline approaching in 2 days, we are eager to address any remaining issues and ensure the paper meets the highest standards.
>
> Your insights are invaluable to us, and we greatly appreciate your time and consideration. Please feel free to share any thoughts you may have.
>
> Looking forward to hearing from you!
>
> Best regards,
>
> Authors

---

### Official Review · Reviewer_asc9 · 2024-11-06

**Soundness:** 2
**Presentation:** 3
**Contribution:** 2
**Rating:** 3
**Confidence:** 5

**Summary:**

This paper introduces LORAM, a training method that refines a pruned model by adjusting the low-rank pruned matrices and then reintegrates dimensionally restored low-rank matrices with the original model for inference. This approach greatly reduces memory demands from model parameters during training and enhances performance by utilizing all original parameters during inference. Consequently, LORAM effectively improves performance on devices with limited memory capacity. Extensive experiments across diverse pruning techniques, model sizes, and task domains demonstrate LORAM's effectiveness.

**Strengths:**

1. The paper is well-written and comprehensively analyzes various adaptations of the proposed method to accommodate different pruning techniques.
2. This paper conducted thorough experiments across various pruning algorithms, models of different sizes, and tasks.

**Weaknesses:**

1. The novelty of this paper is limited as the proposed approach essentially combines existing pruning techniques with LoRA.
2. The claim that LoRAM substantially reduces the number of trainable parameters compared to standard LoRA is somewhat misleading. In the case of unstructured or semi-unstructured pruned models, there is no actual reduction in trainable parameters, as noted in the paper. Meanwhile, the reduction in trainable parameters when fine-tuning structured pruned models is due to the smaller dimensions of these pruned models compared to the original model, rather than any change in the LoRA component, which remains the same as in standard LoRA.
3. Could you explain why, after fine-tuning, structured pruned models outperform semi-structured and unstructured pruned models? In previous studies, like SparseGPT, Wanda, and LLM-pruned, unstructured pruning has consistently shown the least accuracy degradation. However, in your case, the results seem to be the opposite after finetuning.

**Questions:**

Please refer to Weaknesses.

---

> ### Author Response · Authors · 2024-11-20
> **Thanks & Initial Response to Reviewer asc9 (W1)**
>
> **[W1-The novelty of this paper is limited as the proposed approach essentially combines existing pruning techniques with LoRA.]**
>
> While novelty is a multifaceted concept in academic research, we believe it can be broadly viewed from two perspectives:
>
> **(1) Empirical Innovation**: Uncovering new model behaviors and principles, as seen in works like **Scaling Laws** [1] and the **Lottery Ticket Hypothesis** [2], which expand our understanding of the field.
>
> **(2) Technical Innovation**: Developing novel methods or optimizations to advance existing technologies. For instance, **LoRA** [3] enables efficient fine-tuning for large models with low resource cost.
>
> Our contributions on empirical innovation:
>
> **(1) LoRA's Memory Boundaries Have Surplus** : Existing works have aimed to make LoRA more memory-efficient [4,5,6,7], yet the full base model's memory load remains a bottleneck. Our approach reveals that these memory costs can be significantly reduced—by 7 to 8 ×—while maintaining performance gains, as demonstrated on a 70B model across varied datasets, effectively addressing a critical resource efficiency limitation.
>
> **(2) Easily Recoverable Knowledge from Pruning** : Knowledge lost due to pruning can be easily recovered through a low-cost, single-phase continual pretraining process. This alignment can be performed by the model publisher and provides generalizable benefits across different instruction fine-tuning corpora and downstream tasks.
>
> Our contributions on technical innovation:
>
> Our method builds on LoRA but introduces a **unique twist** using two separate models during training and inference, achieving significant memory reduction on training while preserving inference performance. We intentionally keep the method **as simple as possible**, which, in our view, is more of an advantage than a drawback. This simplicity highlights the **unexplored properties** of low-rank adaption, and these insights, along with the new pathways they open for memory-efficient large-model tuning, represent the true innovation of our work.
>
> In summary , our contributions are validated across models, datasets, and tasks, emphasizing the broad applicability and potential impact of our work.
>
> [1] Training compute-optimal large language models.
>
> [2] The Lottery Ticket Hypothesis: Finding Sparse, Trainable Neural Networks.
>
> [3] LoRA: Low-Rank Adaptation of Large Language Models.
>
> [4] LoRA-drop: Efficient LoRA Parameter Pruning Based on Output Evaluation.
>
> [5] LoRA-FA: Memory-efficient Low-Rank Adaptation for Large Language Models Fine-tuning.
>
> [6] Prolora: Partial Rotation Empowers More Parameter-efficient LoRA.
>
> [7] VeRA: Vector-based Random Matrix Adaptation.

---

> ### Author Response · Authors · 2024-11-20
> **Thanks & Initial Response to Reviewer asc9 (W2)**
>
> **[W2-The claim that LoRAM substantially reduces the number of trainable parameters compared to standard LoRA is somewhat misleading. In the case of unstructured or semi-unstructured pruned models, there is no actual reduction in trainable parameters, as noted in the paper. Meanwhile, the reduction in trainable parameters when fine-tuning structured pruned models is due to the smaller dimensions of these pruned models compared to the original model, rather than any change in the LoRA component, which remains the same as in standard LoRA.]**
>
> First, we would like to clarify that our focus is on reducing the memory footprint of the base model during the low-rank training process, as it dominates memory usage compared to the low-rank matrices being trained. We further address the issue from two key perspectives:
>
> 1. **Generality Across Pruning Types**:
>    Our evaluation of various pruning strategies demonstrates the versatility of LoRAM across structured, semi-structured, and unstructured pruning methods. Among these, **structured pruning** is the preferred approach as it physically removes parameters, leading to significant and observable memory reductions. This explains why the majority of our experiments focus on structured pruning, which consistently shows effectiveness across models, datasets, and tasks.
>    In contrast, semi-structured and unstructured pruning rely on sparse matrix optimizations, e.g., **CUTLASS** (CUDA Template for Linear Algebra Subroutines), to achieve practical memory and computational benefits. While these approaches are theoretically viable and may lead to engineering improvements, **optimizing sparse structures is outside the primary scope of this work**.
> 2. **Memory Bottleneck and LoRA Components**:
>    The primary memory bottleneck in low-rank training arises from the base model parameters, not from the LoRA components themselves. As detailed in Section 3.2, for LLaMA-2-13B, the memory footprint of the quantized base model is **11.5× larger** than that of the low-rank matrices, and **46× larger** without quantization. Therefore, **adjusting the LoRA components alone cannot address this bottleneck**. The significant reduction in trainable parameters achieved through structured pruning complements LoRAM’s goal of improving memory efficiency.
>
> These points clarify the rationale behind our approach, underscoring the **simple yet effective advantages of structured pruning** in LoRAM's design.

---

> ### Author Response · Authors · 2024-11-20
> **Thanks & Initial Response to Reviewer asc9 (W3)**
>
> **[W3-Could you explain why, after fine-tuning, structured pruned models outperform semi-structured and unstructured pruned models? In previous studies, like SparseGPT, Wanda, and LLM-pruned, unstructured pruning has consistently shown the least accuracy degradation. However, in your case, the results seem to be the opposite after finetuning.]**
>
> We address the reviewer’s question by analyzing performance at two stages: **after fine-tuning but before recovery**, and **after both fine-tuning and recovery**.
>
> **1. After Fine-Tuning but Before Recovery**:
> In this stage, the results of LoRAM align with prior work (e.g., SparseGPT, Wanda, and LLM-Pruner). **Unstructured and semi-structured pruning consistently outperform structured pruning** (see Figure 6, solid lines). This trend holds true across both aligned and unaligned settings, with **LoRAM-Semi < LoRAM-Unst < LoRAM-Stru < LoRAM-Rand**. The slight advantage of **LoRAM-Semi over LoRAM-Unst** can be attributed to its smaller pruning ratio, which retains more parameters and mitigates performance degradation.
>
> **2. After Fine-Tuning and Recovery**:
> Post-recovery results show that **structured pruning outperforms unstructured pruning**. This can be explained by two factors:
>
> - **Preserved Structure for Recovery**: Structured pruning maintains the organization of the pruned weights into coherent structures (e.g., rows and columns in MLP layers, attention heads in attention layers), ensuring that activations after recovery are aligned with those of the original model. This alignment improves the recovery process.
> - **Pruned Weight Quality**: The quality of pruned weights influences the recovery effectiveness. Structured pruning tends to remove less critical weights, leaving more recoverable parameters. In contrast, unstructured pruning can remove weights that are more difficult to recover, which negatively impacts performance post-recovery.
>
> These results highlight the interplay between pruning strategy and recovery dynamics, suggesting that structured pruning, despite initial performance disadvantages, facilitates more effective recovery. We will incorporate **this discussion in the revised manuscript**.

---

> ### Author Response · Authors · 2024-11-22
> **To Sum Up of Response to Reviewer asc9**
>
> Given the significant impact of your evaluation on our paper and the time constraints of the rebuttal period, we have summarized our responses to facilitate further discussion:
>
> **1. Clarification of Empirical and Technical Novelty (Response to W1):**
>
> - Empirically, we identified and highlighted LoRA's memory surplus during training and the surprising recoverability of pruned weights.
> - Building on these observations, we proposed a technically innovative framework that leverages two distinct models: a small model during training to significantly reduce memory consumption and a full model during inference to ensure high performance. This unique dual-model strategy underpins the practicality and efficiency of LoRAM.
>
> **2. Addressing Concerns Regarding Pruning Strategies (Response to W2):**
>
> - We validated various pruning strategies to demonstrate LoRAM's generalizability. While non-structured pruning can theoretically yield memory savings, it often demands additional engineering efforts for effective implementation.
> - We acknowledge the limitations of modifying LoRA components, as these changes have minimal impact on reducing memory usage during training. Instead, we emphasize the advantages of LoRAM variants with structured pruning, which provide a simple yet effective solution for optimizing the memory consumption of the base model.
>
> **3. Experimental and Analytical Evidence (Response to W3):**
>
> - We provided detailed experimental results and analyses to elucidate the performance differences between unstructured and structured pruning before and after recovery. These findings, we believe, effectively address your concerns and reinforce the robustness of LoRAM.
>
> Given that the concerns you raised have been addressed in detail, we respectfully hope that you will engage in a thorough discussion regarding our work. Furthermore, we kindly request that you reconsider your rating in light of the responses and evidence we have provided. Should you have any additional questions or require further clarification, please do not hesitate to let us know.

---

> ### Author Response · Authors · 2024-11-25
> **Follow-Up: Seeking Further Feedback (Discussion Deadline Approaching in 2 Days)**
>
> Dear Reviewer asc9,
>
> We hope this message finds you well. Following up on our recent exchange regarding this paper, we wanted to kindly check if there are any further concerns or feedback from you. With the discussion deadline approaching in 2 days, we are eager to address any remaining issues and ensure the paper meets the highest standards.
>
> Your insights are invaluable to us, and we greatly appreciate your time and consideration. Please feel free to share any thoughts you may have.
>
> Looking forward to hearing from you!
>
> Best regards,
>
> Authors

---

> > ### Comment · Reviewer_asc9 · 2024-11-27
> >
> > Thank you for the detailed explanation, but I still find the contribution of this work to be marginal. As you mentioned in your response, the primary memory bottleneck in low-rank training stems from the base model parameters rather than the LoRA components. Consequently, the contribution of your work seems limited to the proposal of applying LoRA on top of compressed models that have already been pruned using existing methods to reduce base model parameters. Additionally, the proposed "Pruned Full-Rank Weight Alignment" essentially resembles a commonly used knowledge distillation loss, aligning the outputs of the teacher and student models, without any specific design tailored to address the loss introduced by sparsity in the base model. As such, I will maintain my current score.

---

> > > ### Comment · Reviewer_asc9 · 2024-11-27
> > >
> > > I forgot to mention that using LoRA to recover sparsity loss (pruning loss) has already been extensively adopted in various pruning-related papers, such as "LLM-Pruner: On the Structural Pruning of Large Language Models" and "A Simple and Effective Pruning Approach for Large Language Models" (Wanda). Therefore, I can't entirely agree with your claims that "We are the first to target the memory bottleneck of the base model during low-rank training" and "We are the first to propose a novel pruning-recovery paradigm for LoRA."

---

> ### Author Response · Authors · 2024-11-27
> **Follow up Response to Reviewer asc9**
>
> Thank you for taking the time to provide additional feedback on our clarification.
>
> We would like to **reaffirm the key contributions** of our work, which were recognized during the disscussion period as significant and impactful by our peers. At the outset, we would like to emphasize that the **simplicity and effectiveness** of our approach are core strengths. We believe that introducing unnecessary complexity is not necessary, and keeping things straightforward helps maintain clarity and impact. Our method directly addresses **real-world challenges** faced by resource-constrained users in task-specific customization, offering a solution that is both simple and effective.
>
> Specifically:
>
> 1. **We are the first to target the memory bottleneck of the base model during low-rank training.** While existing memory-efficient LoRA algorithms primarily focus on LoRA components, we identify and tackle the memory overhead of the base model itself—a critical yet often overlooked issue.
> 2. **We are the first to propose a novel pruning-recovery paradigm for LoRA.** By deploying a pruned base model during training and recovering the original full model during inference, our approach achieves up to an **8× reduction in training memory usage**, while preserving the performance gains of full-rank LoRA, even for large models like **LLaMA-3.1-70B**.
> 3. **We are the first to address knowledge inconsistency between the pruned and full models.** Through continued pretraining on the pruned model, we effectively resolve this inconsistency, allowing us to use **high pruning ratios** while maintaining downstream task performance.
> 4. **We demonstrate LoRAM's general effectiveness across diverse scenarios.** Our extensive experiments validate LoRAM's utility across various pruning algorithms, model scales, and downstream tasks. We also provide comprehensive ablation studies that uncover and explain its unique characteristics.
>
> In summary, our work addresses a **critical and previously neglected question**: Can the memory usage of the base model in low-rank training be reduced? We answer this question by introducing an **innovative pruning-recovery paradigm** within the **LoRAM framework** and rigorously validating its effectiveness. Our contributions are threefold: identifying the overlooked bottleneck, proposing a **simple yet impactful solution**, and demonstrating its broad applicability across a range of settings.
>
> We hope this response further highlights the **novelty and significance** of our contributions in addressing a pressing challenge for the community. Thank you once again for your thoughtful review, and we sincerely welcome any additional feedback.
>
> Furthermore, we fully acknowledge your valuable comments.
> If you **agree with the above statements** regarding our novel contributions—such as the phenomena we first uncover, the unique paradigm we propose, and the significant results we present—and have any **practical suggestions** for how these points can be further highlighted or presented in the paper, we would be more than happy to make adjustments. This would greatly help in refining our work!

---

> ### Author Response · Authors · 2024-11-27
> **Follow up Response to Reviewer asc9**
>
> Thank you for engaging in a detailed discussion about the contributions of LoRAM, which is crucial for understanding the significance of our work. We appreciate your feedback and would like to clarify the two statements in question:
>
> **1. "We are the first to target the memory bottleneck of the base model during low-rank training"**
>
>    While existing pruning methods such as LLM-Pruner and Wanda focus on creating high-quality compressed models for downstream tasks, their primary goal is not to address the memory bottleneck during LoRA training. Our work uniquely targets this bottleneck, where the memory consumption of the base model is a critical challenge. Existing methods struggle at high pruning rates—for instance, LLM-Pruner is typically effective only up to 10%-20% pruning, which is insufficient for significantly reducing memory usage. In contrast, LoRAM achieves up to an 8× reduction in trainable parameters while maintaining downstream task performance. This distinct focus sets our contribution apart.
>
> **2. "We are the first to propose a novel pruning-recovery paradigm for LoRA"**
>
>    The novelty of our pruning-recovery paradigm lies in training low-rank matrices on a pruned base model to drastically reduce training memory, followed by recovering these matrices to full dimensions during inference to ensure competitive generation quality. This is fundamentally different from existing pruning methods, such as LLM-Pruner, which train and infer directly on pruned models (Essentially a variation of LoRAM (w/o Recovery)). Even with post-training training, these approaches typically handle only modest pruning ratios (e.g., 10%-20%). LoRAM's paradigm overcomes these limitations, enabling substantial memory savings during training without compromising inference quality.
>
> In summary, as emphasized in our introduction, achieving memory-efficient LoRA training with significant performance gains is challenging due to the limitations of directly applying existing pruning methods at high pruning rates. However, our extensive experiments demonstrate that these methods are **undoubtedly an effective 'lever'** when **combined with the novel low-rank recovery introduced in LoRAM**.
> We hope this clarifies our contributions and addresses your concerns. Thank you again for your valuable feedback!

---

### Official Review · Reviewer_aWyt · 2024-11-06

**Soundness:** 3
**Presentation:** 3
**Contribution:** 3
**Rating:** 6
**Confidence:** 4

**Summary:**

This paper proposes “LoRAM,” a novel memory-efficient LoRA fine-tuning process that significantly reduces required storage by pruning base weights—an approach largely overlooked in prior LoRA research. To mitigate the discrepancy introduced by pruning, the authors introduce a recovery step to ensure alignment at inference time, thereby improving accuracy.

Compared to models with a similar effective parameter reduction ratio, LoRAM achieves a higher compression rate while demonstrating lower training/test loss and higher downstream accuracy on the LLaMA-2 family and LLaMA-3.1 models. Additionally, an ablation study shows that the proposed recovery and alignment steps contribute to improved training performance.

**Strengths:**

This paper introduces a novel approach to optimizing the heavy base weights in LoRA-based fine-tuning, an area not well-focused in existing LoRA studies. Moving beyond the widely studied quantization, the authors address the challenges of achieving high compression through pruning by introducing recovery and alignment steps, which help to offset pruning’s drawbacks and yield promising performance results. The method shows scalability potential, and the presentation and explanation are clear, making it easier to follow what might otherwise be a complex process. The experiments also appear to be thoroughly conducted.

**Weaknesses:**

- The paper lacks a discussion of the cost implications of the proposed method. Unlike standard one-stage LoRA fine-tuning, the multi-stage process in LoRAM may involve trade-offs in latency, but this is not adequately addressed. Given the focus on efficient training, a detailed comparison of memory and latency costs with baseline methods should be discussed.

- While the main table’s comparisons by compression ratio are understandable, the results should also include fine-tuning outcomes on same LoRAM’s target LLM, as seen in Figure 8. Including the performance upper bound of fine-tuning the same model without memory reduction would allow readers to better assess the trade-off between memory compression and performance.

- To elaborate on the need for comparison within the same model family: from the perspective of a model publisher, different sizes of LLMs are not just scaled versions but may be trained with distinct capabilities in mind, considering future usability (ex. LLaMA-3.1-8/70B vs. LLaMA-3.2-3B). Thus, comparing models with similar reduction ratios but different original capabilities may not be appropriate. For those who performing fine-tuning, it would be more informative to see how LoRAM compares to standard LoRA on the same model, as this would directly reflect the practical value of the method. (I’m open to counterarguments if there’s a rationale for this choice.)

**Questions:**

- Was a comparable grid search conducted for hyper-parameter tuning (learning rate, epochs) for both baseline LoRA and LoRAM? Fine-tuning can be sensitive to settings, so the degree of granularity in hyper-parameter tuning might impact the final performance.
- I am curious about whether this method would also yield effective results for domain-specific fine-tuning tasks.

---

> ### Author Response · Authors · 2024-11-20
> **Thanks & Initial Response to Reviewer aWyt (W1)**
>
> **[W1-The paper lacks a discussion of the cost implications of the proposed method. Unlike standard one-stage LoRA fine-tuning, the multi-stage process in LoRAM may involve trade-offs in latency, but this is not adequately addressed. Given the focus on efficient training, a detailed comparison of memory and latency costs with baseline methods should be discussed.]**
>
> Identifying the costs of LoRAM is indeed important, which is why we report both the **number of training tokens** used during the alignment phase and the **parameter reduction ratios** in the low-rank training phase. We address the reviewer’s concern with the following clarifications:
>
> LoRAM comprises two stages: an **offline alignment phase** and an **online low-rank matrix** **training phase**.
>
> **1. Offline Alignment Phase (Task-Agnostic):**
>
> The offline phase is task-agnostic and can be conducted by the model publisher prior to deployment, making its cost negligible for end users. To quantify the offline cost, we measured the **number of training tokens** (as in [1]) rather than end-to-end latency, which can **vary** based on **hardware configurations** . As shown in Figure 5, LoRAM achieves significant performance gains using only 13 million tokens, demonstrating the efficiency of the alignment phase.
>
> **2. Online Low-Rank Matrix Training Phase:**
>
> For the online phase, the **memory and latency costs** are **primarily determined** by the **size of the base model parameters**, which dominate resource consumption during training. To avoid redundancy in reporting, we focused on **parameter reduction ratios** instead of absolute time or memory usage.
>
> We recognize the value of providing additional metrics and include memory and latency comparisons for online training in follow table to directly address this concern. We conducted additional experiments with a workload of 1024 samples (batch size 128, micro-batch size 4, sequence length 512) randomly selected from OpenHermes. The results, summarized in the table below, show that **LoRAM with a structured pruning ratio of 2.17× (13B → 6B)** achieves comparable peak memory, latency, and throughput to **7B LoRA**, with only minor trade-offs. These minor differences arise from the larger layer count in 13B LoRAM, which introduces more non-GEMM operations, slightly affecting latency and throughput.
>
> These results demonstrate the **simple-yet-effective advantages** of LoRAM's design in achieving substantial resource efficiency without significant trade-offs in memory or latency.
>
> [1] Xia, M., Gao, T., Zeng, Z., & Chen, D. Sheared LLAMA: Accelerating Language Model Pre-training via Structured Pruning. ICLR 2024.
>
> |                        | Parameters Size of Base Model | Parameter Reduction Ratio | Peak Memory (MiB) | Latency (s) | Throughput (sample/s) |
> | ---------------------- | ----------------------------- | ------------------------- | ----------------- | ----------- | --------------------- |
> | LLaMA-2-7B LoRA        | 6738415651                    | 1.93×                     | 30517             | **134.27**  | **7.626**             |
> | LLaMA-2-13B LoRA       | 13015864320                   | 1.00×                     | 51661             | 206.07      | 4.969                 |
> | LLaMA-2-13B LoRAM-Stru | **6005662720**                | **2.17×**                 | **29799**         | 147.86      | 6.925                 |

---

> ### Author Response · Authors · 2024-11-20
> **Thanks & Initial Response to Reviewer aWyt (W2)**
>
> **[W2-While the main table’s comparisons by compression ratio are understandable, the results should also include fine-tuning outcomes on same LoRAM’s target LLM, as seen in Figure 8. Including the performance upper bound of fine-tuning the same model without memory reduction would allow readers to better assess the trade-off between memory compression and performance.]**
>
> We agree that fine-tuning the same model without parameter reduction represents the upper bound of our method.
> However, we omitted this from the main table to avoid redundancy, as the upper bound is evident and implicitly reflected in related comparisons. For example, in Table 1, we included core competitive baselines such as **7B LoRA and 13B w/o FT for the 13B model**. Adding the 13B LoRA upper bound would **duplicate information** already represented in the lower section of Table 1 for the 70B model, where 13B LoRA is used as a baseline.
>
> To better highlight the trade-off between memory compression and performance, we will include these upper-bound results (primarily for **LLaMA-2-70B LoRA**) in the appendix of the revised version and explicitly reference them in the main text. As an example, the following CSR benchmark results illustrate this trade-off for **LLaMA-2-13B**:
>
> | Model          | CSR （Averge Accuracy % ） | Parameter Reduction Ratio |
> | -------------- | --------------------- | ------------------------- |
> | 7B LoRA        | 61.51±1.29           | 1.93×                    |
> | 13B w/o FT     | 64.28±1.30           | 1.00×                    |
> | 13B LoRAM-Rand | 64.64±1.29           | **2.17×**          |
> | 13B LoRA | **65.05±1.29** | 1.00× |
>
> This addition will provide greater clarity regarding the upper bound and further contextualize the benefits of memory compression achieved by LoRAM.

---

> ### Author Response · Authors · 2024-11-20
> **Thanks & Initial Response to Reviewer aWyt (W3)**
>
> **[W3-To elaborate on the need for comparison within the same model family: from the perspective of a model publisher, different sizes of LLMs are not just scaled versions but may be trained with distinct capabilities in mind, considering future usability (ex. LLaMA-3.1-8/70B vs. LLaMA-3.2-3B).  Thus, comparing models with similar reduction ratios but different original capabilities may not be appropriate. For those who performing fine-tuning, it would be more informative to see how LoRAM compares
> to standard LoRA on the same model, as this would directly reflect the practical value of the method. (I’m open to counterarguments if there’s a rationale for this choice.]**
>
> We appreciate the reviewer’s concern and would like to clarify our rationale for comparing models of different scales within the same series.
>
> **1. Practical Considerations for Fine-Tuning:** Our comparison reflects practical use cases where resource constraints often guide model selection. Practitioners typically choose between **deploying larger models with reduced parameters (e.g., through LoRAM) or smaller models with full parameter usage**. By including smaller models in the comparison, we aim to illustrate the trade-offs and advantages LoRAM offers in resource-constrained scenarios.
>
> **2. Same-Series Comparisons Ensure Consistency:** We focus on models within the same series (e.g., LLaMA-2 or LLaMA-3) to ensure consistent training paradigms, architectural similarities, and comparable baselines. While larger and smaller models in the same series may not be scaled versions in a strict sense, they **share foundational design and pretraining data**. This consistency minimizes confounding factors unrelated to scale, such as divergent pretraining objectives or domain-specific customizations.
>
> **3. Performance Gains Relative to w/o FT Baseline:**
> For fine-tuning users, LoRAM shows **consistent performance improvements** over the baseline without fine-tuning (w/o FT). For example, on LLaMA-3.1-70B, LoRAM achieves notable perplexity (PPL) reductions using QLoRAM while outperforming the w/o FT baseline across downstream tasks. These results validate LoRAM’s robustness **across model size scales**, demonstrating its practical benefits for resource-constrained scenarios.
>
> **4. Competitiveness with Standard LoRA:**
> When compared directly with standard LoRA, LoRAM maintains strong competitiveness. For instance, on LLaMA-3.1-70B with QLoRAM-Stru 400, LoRAM performs comparably or better on **downstream tasks like MathQA and CSR**, while significantly reducing memory requirements. This highlights its efficiency and utility in delivering comparable performance at a lower resource cost.

---

> ### Author Response · Authors · 2024-11-20
> **Thanks & Initial Response to Reviewer aWyt (Q1,Q2)**
>
> **[Q1-Was a comparable grid search conducted for hyper-parameter tuning (learning rate, epochs) for both baseline LoRA and LoRAM? Fine-tuning can be sensitive to settings, so the degree of granularity in hyper-parameter tuning might impact the final performance.]**
>
> Yes, we conducted a comparable grid search for hyper-parameter tuning, including learning rates and epochs, across all baselines, including LoRA and LoRAM. The search range for learning rates was [1e-5, 3e-3].
>
> This process is reflected in Figures 3, 4, and 5, where LoRA-trained models of the same scale consistently achieve the upper bound in perplexity (PPL), demonstrating that the results are not constrained by insufficient tuning. Moreover, the performance gap between LoRA and LoRAM remains significant, even with comprehensive hyper-parameter optimization.
>
> In the revised version of [**Appendix F**](https://openreview.net/pdf?id=s7DkcgpRxL), we have added details on the learning rate tuning process for full LoRA on LLaMA-2-7B and LLaMA-2-13B models using the OpenHermes dataset. These experiments show that a learning rate of **1e-3 consistently delivers the best perplexity across both in-domain and out-of-domain datasets**, further supporting the validity of our comparison.
>
>
> **[Q2-I am curious about whether this method would also yield effective results for domain-specific fine-tuning tasks.]**
>
> Our method has been evaluated on general instruction fine-tuning datasets, achieving performance improvements across nine diverse downstream tasks, which represent a more challenging and comprehensive scenario.
>
> We acknowledge the value of domain-specific fine-tuning tasks and are actively exploring this direction. We will share the results as soon as they become available.
>
> ----
> ### **Update 11.23 (Peformance of Domain-Specific Task):**
>
> We are excited to share our **latest results on domain-specific fine-tuning tasks**. We evaluated LoRAM on the GSM8K dataset's training set, a challenging mathematical reasoning benchmark that is highly sensitive to sparsification. Specifically, we trained LLaMA-3.1-70B using QLoRAM.
>
> As demonstrated in the table below, LoRAM maintains **strong performance even in such domain-specific tasks**. These results effectively address concerns about the applicability of LoRAM in specialized domains. This experiment has been added to  [**Appendix G**](https://openreview.net/pdf?id=s7DkcgpRxL), in the revised version of the paper to highlight the method's effectiveness further.
>
> | **LLaMA-3.1 Model**                      | **GSM8K Accuracy (%)** | **Parameter Reduction Ratio** |
> | ---------------------------------------------- | ---------------------------- | ----------------------------------- |
> | **8B w/o FT**                            | 55.27                        | **8.79×**                    |
> | **8B LoRA (OpenHermes 400)**             | 55.80                        | **8.79×**                    |
> | **70B w/o FT**                           | 75.28                        | 1.00×                              |
> | **70B QLoRAM-Stru 400 (OpenHermes 400)** | 80.36                        | 7.07×                              |
> | **70B QLoRAM-Stru 400 (GSM8K 100)**      | 77.18                        | 7.07×                              |
> | **70B QLoRAM-Stru 400 (GSM8K 200)**      | 79.15                        | 7.07×                              |
> | **70B LoRA (OpenHermes 400)**            | **80.74**              | 1.00×                              |

---

> > ### Comment · Reviewer_aWyt · 2024-11-24
> >
> > Thank you for the detailed response and experimental results. The authors’ answers have addressed most of my concerns. I believe this work presents a novel approach (train small, infer large) by leveraging both pruning and quantization simultaneously, offering a direction for achieving efficient yet high-performing LLMs. However, further investigation is needed into the potential degradation (not revealed by academic benchmarks) that might occur when offline alignment is applied by model publishers for additional efficiency. That being said, I am happy to increase my score.

---

> > > ### Author Response · Authors · 2024-11-24
> > > **Follow up Response to Reviewer aWyt**
> > >
> > > Thank you for recognizing our article's novelty, importance, and presentation. We greatly appreciate your increase in the score based on our response, as it is crucial for our work!

---

### Official Review · Reviewer_EtZ8 · 2024-11-09

**Soundness:** 3
**Presentation:** 1
**Contribution:** 3
**Rating:** 8
**Confidence:** 4

**Summary:**

The paper proposes LORAM, a memory-efficient training approach for Low-Rank Adaptation (LoRA) fine-tuning on large language models (LLMs). LORAM aims to reduce the memory burden during training by pruning the model parameters and then “recovering” them during inference. The technique includes an offline alignment process to minimize knowledge discrepancies between the pruned and original models, as well as integration with quantization to further optimize memory usage. Extensive experiments showcase LORAM’s performance across various tasks using models like LLaMA-2.

**Strengths:**

- LORAM aims to make LLM fine-tuning feasible on lower-memory devices, addressing a relevant challenge for the NLP community.
- The introduction of an alignment phase to handle pruning inconsistencies is conceptually innovative and could provide insights for future memory-efficient models.
- The authors conduct a broad range of experiments, which helps to illustrate LORAM’s performance across multiple tasks.

**Weaknesses:**

- The paper contains frequent grammatical errors, inconsistent terminology, and formatting issues that detract from readability.
- The main technical idea—pruning during training and recovering for inference—lacks fundamental novelty and could be seen as an incremental step on existing methods rather than a breakthrough.
- The experimental setup would be more convincing with comparisons to simpler or standard pruning approaches, helping to illustrate LORAM’s distinct benefits more clearly.
- Testing LORAM beyond LLaMA models would make the approach more broadly applicable and better showcase its robustness across architectures.

**Questions:**

1. Could the authors provide specific examples of how LORAM compares to standard LoRA without any pruning? Same goes for QLORAM vs QLORA?
2. What influenced the choice of pruning ratios, and could different ratios impact the effectiveness of LORAM’s alignment step? Could the authors also clarify if any theoretical or empirical basis underlies the decision to use certain pruning ratios in LORAM? For instance, what drives the choice of an optimal parameter reduction ratio?
3. Are there cases where LORAM underperforms compared to standard LoRA? This would provide insights into LORAM’s limitations. As it is, I find it hard to wrap my head around the possibilities of LORAM winning in all scenarios.
4. How was the the alignment corpus chosen, and what effect does corpus size have on the performance of aligned models?
5. How consistent are LORAM’s improvements across different tasks? Further analysis here would help illustrate LORAM’s potential.

---

> ### Author Response · Authors · 2024-11-18
> **Thanks & Initial Response to Reviewer EtZ8 (W1,W2)**
>
> **[W1-The paper contains frequent grammatical errors, inconsistent terminology, and formatting issues that detract from readability.]**
>
> We appreciate the reviewer’s feedback on language and formatting issues. We will thoroughly proofread and revise the paper to address these concerns. Any specific examples the reviewer could provide would be greatly appreciated and help us improve the paper more efficiently.
>
> **[W2-The main technical idea—pruning during training and recovering for inference—lacks fundamental novelty and could be seen as an incremental step on existing methods rather than a breakthrough.]**
>
> While novelty is a multifaceted concept in academic research, we believe it can be broadly viewed from two perspectives:
>
> **(1) Empirical Innovation**: Uncovering new model behaviors and principles, as seen in works like **Scaling Laws** [1] and the **Lottery Ticket Hypothesis** [2], which expand our understanding of the field.
>
> **(2) Technical Innovation**: Developing novel methods or optimizations to advance existing technologies. For instance, **LoRA** [3] enables efficient fine-tuning for large models with low resource cost.
>
> Our contributions on empirical innovation:
>
> **(1) LoRA's Memory Boundaries Have Surplus** : Existing works have aimed to make LoRA more memory-efficient [4,5,6,7], yet the full base model's memory load remains a bottleneck. Our approach reveals that these memory costs can be significantly reduced—by 7 to 8 ×—while maintaining performance gains, as demonstrated on a 70B model across varied datasets, effectively addressing a critical resource efficiency limitation.
>
> **(2) Easily Recoverable Knowledge from Pruning** : Knowledge lost due to pruning can be easily recovered through a low-cost, single-phase continual pretraining process. This alignment can be performed by the model publisher and provides generalizable benefits across different instruction fine-tuning corpora and downstream tasks.
>
> Our contributions on technical innovation:
>
> Our method builds on LoRA but introduces a **unique twist** using two separate models during training and inference, achieving significant memory reduction on training while preserving inference performance. We intentionally keep the method **as simple as possible**, which, in our view, is more of an advantage than a drawback. This simplicity highlights the **unexplored properties** of low-rank adaption, and these insights, along with the new pathways they open for memory-efficient large-model tuning, represent the true innovation of our work.
>
> In summary , our contributions are validated across models, datasets, and tasks, emphasizing the broad applicability and potential impact of our work.
>
> [1] Training compute-optimal large language models.
>
> [2] The Lottery Ticket Hypothesis: Finding Sparse, Trainable Neural Networks.
>
> [3] LoRA: Low-Rank Adaptation of Large Language Models.
>
> [4] LoRA-drop: Efficient LoRA Parameter Pruning Based on Output Evaluation.
>
> [5] LoRA-FA: Memory-efficient Low-Rank Adaptation for Large Language Models Fine-tuning.
>
> [6] Prolora: Partial Rotation Empowers More Parameter-efficient Lora.
>
> [7] VeRA: Vector-based Random Matrix Adaptation.

---

> ### Author Response · Authors · 2024-11-18
> **Thanks & Initial Response to Reviewer EtZ8 (W3,W4)**
>
> **[W3-The experimental setup would be more convincing with comparisons to simpler or standard pruning approaches, helping to illustrate LoRAM's distinct benefits more clearly.]**
>
> Discussing LoRAM in the context of simple and standard pruning methods is indeed crucial to demonstrating its unique advantages. We will address this in two parts: first, by illustrating LoRAM's effectiveness when combined with simple pruning methods; second, by highlighting the limitations of naive pruning when paired with LoRA.
>
> **(1) Effectiveness of Simple & Standard Pruning:** In our experiments, we validated the effectiveness of LoRAM under **random structured pruning** (LoRAM-Rand) across various model families, fine-tuning datasets, and downstream tasks. Random pruning, as a widely-used and sufficiently simple baseline, demonstrated robust performance while revealing LoRAM's **unique advantages**. Building on this baseline, we further explored **diverse pruning schemes**, including semi-structured and unstructured pruning, which underscored LoRAM's **generality**. The **low sensitivity to pruning criteria** can be attributed to the high pruning ratios (e.g., >65%), which tend to reduce performance differences between pruning strategies.
>
> **(2) Limitations of Naive Pruning with LoRA:** As shown in Figure 6 of the submission, **naive pruning** followed by direct **LoRA** fine-tuning (w/o Recovery and Alignment) yields limited performance improvements. This emphasizes the critical role of LoRAM’s recovery and alignment stages in achieving superior results.
>
> We believe these experiments and analyses comprehensively demonstrate LoRAM’s distinct benefits over simpler pruning approaches, reinforcing its practical value and broader applicability.
>
> **[W4-Testing LoRAM beyond LLaMA models would make the approach more broadly applicable and better showcase its robustness across architectures.]**
>
> We agree that testing LoRAM on additional models would enhance its applicability and better showcase its robustness. This is why we evaluated both LLaMA-2 and LLaMA-3.1. Our focus on LLaMA is driven by two key reasons:
>
> **(1) Widespread Adoption and Maintenance:** LLaMA is one of the most widely adopted and actively maintained open-source LLM families, ensuring that LoRAM’s effectiveness on these models is relevant to the broader research community. Notably, LoRAM also demonstrates superior performance on the latest LLaMA-3.1.
>
> **(2) Pruning algorithm-friendly:** LLaMA models serve as the primary architectures for evaluating various pruning methods (e.g., LLM-Pruner and SparseGPT), showcasing their inherent compatibility with diverse pruning techniques. This choice also facilitates community tracking and reproducibility.
>
> While our initial experiments have focused on LLaMA, we are actively exploring LoRAM’s adaptation to other architectures and aim to include updated results as we progress.

---

> ### Author Response · Authors · 2024-11-18
> **Thanks & Initial Response to Reviewer EtZ8 (Q1)**
>
> **[Q1-Could the authors provide specific examples of how LoRAM compares to standard LoRA without any pruning? Same goes for QLoRAM vs QLoRA?]**
>
> Our submission includes comparisons of LoRAM against standard LoRA without pruning at the same model scale to highlight their respective trade-offs between performance and parameter efficiency.
>
> As shown in Figure 4, at comparable scales (e.g., LLaMA-2-13B and LLaMA-2-70B) and across fine-tuning datasets such as OpenHermes and OpenOrca, standard LoRA consistently achieves the lowest perplexity (PPL), serving as the performance upper bound. LoRAM, while incurring a slight increase in PPL, achieves significant parameter reductions and outperforms smaller models fine-tuned with LoRA, underscoring its suitability for efficiency-critical scenarios.
>
> While PPL may not fully reflect downstream performance due to data and task distribution differences, the observed trends persist. For example, Table 2 shows that on the CSR benchmark (averaged over six tasks), LoRAM-Rand achieves: **13B LoRA > 13B LoRAM-Rand > 13B w/o FT > 7B LoRA**, with a 2.17× reduction in parameters compared to standard LoRA. To further clarify, we have added detailed OpenHermes results in the rebuttal to illustrate these trends more explicitly.
>
> Regarding QLoRA and QLoRAM, QLoRA provides a memory-efficient alternative to standard LoRA, trading slight performance for reduced memory usage. For example, using LLaMA-3.1-70B, QLoRA achieves approximately a **4× parameter reduction** while maintaining near-equivalent performance to standard LoRA. LoRAM variants extend this principle with additional parameter reductions (e.g., 7.07× for QLoRAM-Stru 400) at a small cost to performance. On the GSM8K benchmark, this translates to the trend: **70B LoRA  ≥ 70B QLoRA > 70B QLoRAM-Rand > 70B w/o FT > 8B LoRA.**
>
> Below are specific examples illustrating these trends:
>
> | Model          | CSR（Average Accuracy %） | Parameter Reduction Ratio |
> | -------------- | --------------------- | ------------------------- |
> | LLaMA-2-7B LoRA | 61.51±1.29           | 1.93×             |
> | LLaMA-2-13B w/o FT | 64.28±1.30           | 1.00×                    |
> | LLaMA-2-13B LoRAM-Rand | 64.64±1.29    | **2.17×**          |
> | LLaMA-2-13B LoRA | **65.05±1.29** | 1.00×                    |
>
> | Model                         | GSM8K     | Parameter Reduction Ratio |
> | ----------------------------- | --------- | ------------------------- |
> | LLaMA-3.1-8B LoRA             | 55.80     | **8.79×**                     |
> | LLaMA-3.1-70B w/o FT          | 75.28     | 1.00×                     |
> | LLaMA-3.1-70B QLoRAM-Stru 400 | 80.36     | 7.07×                 |
> | LLaMA-3.1-70B QLoRA           | ≤80.74    | 4.00×               |
> | LLaMA-3.1-70B LoRA            | **80.74** | 1.00×                     |

---

> ### Author Response · Authors · 2024-11-18
> **Thanks & Initial Response to Reviewer EtZ8 (Q2)**
>
> **[Q2-What influenced the choice of pruning ratios, and could different ratios impact the effectiveness of LoRAM's alignment step? Could the authors also clarify if any theoretical or empirical basis underlies the decision to use certain pruning ratios in LoRAM? For instance, what drives the choice of an optimal parameter reduction ratio?]**
>
> We address the question in three parts for clarity:
>
> **Q2-1: What influenced the choice of pruning ratios?**
>
> The pruning ratios were chosen based on a **core competitive scenario**. For instance, with LLaMA-2-13B, a LoRAM-trained 13B model (parameter reduction ratio: 1.95×~2.17×) should outperform the original untrained model while achieving a parameter reduction close to or slightly lower than that of a LoRA-trained 7B model (1.93×). For LLaMA-2-70B, we observed that even with a parameter reduction ratio as high as **8.21×**, LoRAM still delivers improvements in both perplexity (PPL) and downstream task performance, highlighting its robustness across varying compression levels.
>
> **Q2-2: Could different ratios impact the effectiveness of LoRAM's alignment step?**
>
> Theoretically, smaller pruning ratios require less alignment data. However, for models of scale 13B or larger, we consistently prune over 65% of parameters to maintain alignment with the core competitive scenario. For example, in our experiments with LLaMA-3.1-70B, updating only 13 million tokens during the alignment step suffices to yield competitive performance gains. Figure 7 demonstrates LoRAM’s effectiveness under various pruning ratios, which remains significantly superior to naive pruning.
>
> **Q2-3: What drives the choice of an optimal parameter reduction ratio?**
>
> We did not seek a universally optimal parameter reduction ratio; this choice fundamentally depends on user-specific memory constraints. Instead, we provide a reference competitive scenario, as discussed above. Furthermore, Section 4.6 shows that even with a reduction ratio as high as 8.21×, LoRAM still achieves competitive improvements in both PPL and downstream task performance.

---

> ### Author Response · Authors · 2024-11-18
> **Thanks & Initial Response to Reviewer EtZ8 (Q3,Q4)**
>
> **[Q3-Are there cases where LoRAM underperforms compared to standard LoRA? This would provide insights into LoRAM's limitations. As it is, I find it hard to wrap my head around the possibilities of LoRAM winning in all scenarios.]**
>
> Certainly, LoRAM is designed to trade some performance for significant memory reduction, and thus, it theoretically underperforms standard LoRA at the same model scale, as LoRA serves as the performance upper bound.
>
> Using LLaMA-3.1 as an example (Figure 5), LoRAM’s perplexity (PPL) performance falls between that of 8B LoRA and 70B LoRA. This pattern is consistent in downstream tasks, such as GSM8K:
>
> | Model               | GSM8K           | Parameter Reduction Ratio |
> | ------------------- | --------------- | ------------------------- |
> | 8B w/o FT           | 55.27           | **8.79×**          |
> | 8B LoRA             | 55.80           | **8.79×**          |
> | 70B w/o FT          | 75.28           | 1.00×                    |
> | 70B QLoRAM-Stru 400 | 80.36    | 7.07×             |
> | 70B LoRA            | **80.74** | 1.00×                    |
>
> This table highlights LoRAM’s trade-off: achieving competitive performance while significantly reducing parameter count. Although it may not surpass standard LoRA in absolute performance, its efficiency makes it valuable in resource-constrained scenarios. This feature is extremely appealing for numerous applications. Previously, **memory constraints** hindered them to utilize the superior performance of full base models, even with LoRA. **Now, this can be accomplished with LoRAM.**
>
> **[Q4-How was the the alignment corpus chosen, and what effect does corpus size have on the performance of aligned models?]**
>
> The alignment corpus was chosen from **widely-used**, **open-source** datasets such as FineWeb and OpenMathWeb, without imposing strict selection constraints.
>
> Regarding the impact of corpus size on aligned model performance, Section 4.4 provides a detailed discussion. Our experiments demonstrate that even with a corpus size as small as **13 million** tokens, competitive performance can be achieved, showing that effective results are attainable with smaller corpora. Additionally, as this alignment process can be performed offline by the model provider, the training cost can be **flexibly adjusted to suit specific requirements**.

---

> ### Author Response · Authors · 2024-11-18
> **Thanks & Initial Response to Reviewer EtZ8 (Q5)**
>
> **[Q5-How consistent are LoRAMs improvements across different tasks? Further analysis here would help illustrate LoRAM's potential.]**
>
> We have provided LoRAM's performance across three categories and nine downstream tasks, including sparsity-sensitive domains such as mathematics [1] (e.g., GSM8K). LoRAM demonstrates relatively stable and competitive performance, highlighting its potential utility. For instance, with LLaMA-2-70B, LoRAM consistently outperforms baseline models in several key metrics, as shown in the table below for OpenHermes. Furthermore, Figure 8 illustrates performance trends across various pruning ratios and downstream tasks, offering additional evidence of LoRAM's robustness and addressing potential concerns about its general applicability.
>
> | Model           | MathQA | GSM8K | CSR         | Pass@1 | Pass@10 | Parameter Reduction Ratio |
> | --------------- | ------ | ----- | ----------- | ------ | ------- | ------------------------- |
> | 70B w/o FT      | 39.53  | 52.01 | 68.69±1.27 | 31.71  | 58.54   | 1.00×     |
> | 13B LoRA        | 32.03  | 36.69 | 65.05±1.29 | 18.29  | 35.98   | 1.00×     |
> | 70B QLoRAM-Stru | **39.77** | **57.16** | **69.10±1.27** | **32.32** | **58.54** | **6.27×**              |
>
> [1] Zhou, Y., Chen, Z., Xu, Z., Lin, V., & Chen, B. Sirius: Contextual Sparsity with Correction for Efficient LLMs. arXiv, 2024.

---

> > ### Comment · Reviewer_EtZ8 · 2024-11-23
> >
> > Thanks for significantly answering most of my questions. I have decided to raise my score.

---

> > > ### Author Response · Authors · 2024-11-24
> > > **Follow up Response to Reviewer EtZ8**
> > >
> > > Thank you for your updated evaluation and for raising the score, which affirms the innovation and contributions of our work. We are especially grateful for your recognition of the improvements in our presentation and the value of our approach. Your feedback has been instrumental in enhancing the quality of our manuscript, and we deeply appreciate your support!

---

### Author Response · Authors · 2024-11-28
**General Response**

For clarity and simplicity, we will refer to Reviewers EtZ8, aWyt, asc9, H1mX, and BSMa as R1, R2, R3, R4, and R5, respectively, in the following response.

We sincerely thank all reviewers for their thoughtful and constructive feedback. We are encouraged by their recognition of the key contributions and strengths of our work.

In particular, we sincerely appreciate the recognition of the *importance* of our research problem, which addresses the *critical challenge* in the NLP community of achieving memory-efficient LoRA on resource-constrained devices (**R1, R2, R5**). Specifically, our focus on alleviating the memory bottleneck caused by the base model in LoRA training—an issue *not explored* by existing methods—is *greatly valued* (**R2**).
We are grateful for the recognition of our method’s *simplicity and effectiveness* (**R2, R5**), with the recovery and alignment steps *effectively* mitigating the drawbacks of pruning and offering *valuable insights* for future memory-efficient models (**R1**). Furthermore, LoRAM's potential as a *promising solution* for the field is acknowledged, and its *high compatibility \& scalability* with sparsification and quantization techniques is recognized (**R3, R4, R5**).
Additionally, we appreciate the reviewers’ recognition of our *comprehensive experiments* (**R1, R2, R3**), particularly the evaluation across *different pruning methods* (**R3, R4, R5**), *model scales* (**R2, R3**), and *tasks* (**R1, R2**), which lead to *impressive results* (**R4**).
Finally, we are grateful for the reviewers' feedback highlighting that our paper is *easy to follow* (**R2, R5**), *well-written* (**R3, R4**), and *well-structured* (**R4**), with a *clear presentation* (**R2, R5**). The visualizations have been noted as *quite helpful for quickly grasping the method* (**R4**), and the results analysis is *comprehensive* (**R3**).

We have carefully addressed each individual comment provided by the reviewers.
Below, we summarize the core contributions of our study, the updates to our experiments, and the in-depth discussions in our revision.

------

**Core Contributions of Our Work**

**1. Critical Problem Identification**：We identify a critical issue that existing memory-efficient LoRA approaches overlook but must address: whether the memory overhead of the base model, which dominates the memory usage in LoRA, can be further reduced while maintaining inference accuracy.

**2. Novel Training Scheme**: We propose LoRAM, the first memory-efficient LoRA training scheme based on a pruning-recovery process. LoRAM updates the weights retained through pruning, which significantly reduces memory usage and training time, while employing the pruned weights during inference to enhance generation performance.

**3. Effective Alignment Strategy:** We identify that the knowledge inconsistency between the pruned model used for training and the original model used for inference limits the performance gain of LoRAM under aggressive pruning rates. To address this, we introduce an alignment strategy, training the pruned model on a small amount of general corpus. This one-shot offline process is simple to execute and can be easily performed by the model publisher.

**4. Extensive Experimental Evaluation**: We conduct comprehensive experiments to validate the effectiveness of LoRAM across various pruning algorithms, models of different sizes, and tasks in different domains. Notably, QLoRAM, which combines LoRAM with structured pruning and 4-bit quantization, reduces the memory cost of LLaMA-2-70B parameters by 8.21×, while achieving performance gains superior to both the original LLaMA-2-70B and the LoRA-trained LLaMA-2-13B.

------

**Updates of experimental results during Rebuttal**

- **`Appendix F`**: Added learning rate tuning details for LoRA on LLaMA-2-7B and LLaMA-2-13B using the OpenHermes dataset, showing the fairness of hyperparameter settings and result validity.
- **`Appendix G`**: Added new results on domain-specific fine-tuning tasks, addressing concerns about LoRAM’s applicability in specialized domains.
- **`Appendix H`**: Compared the cost of baselines, including parameters, reduction ratio, peak memory, latency, and throughput.

**Updates of in-depth discussions during Rebuttal**

- **`Appendix C.5`**:  Included findings on unchanged weights to support LoRAM’s motivation with visual, theoretical, and quantitative evidence, as well as scalability.
- **`Appendix I`**:  In-depth analysis of the causes behind the changes in performance trends before and after recovery for LoRAM under various pruning strategies.

------

We believe these additions and clarifications comprehensively address the reviewers' concerns and enhance the overall quality of our manuscript.  All revisions are highlighted in `blue` text for ease of reference. Our manuscript is updated on `Nov 27, AOE time`.

We look forward to the reviewers' favorable consideration and remain grateful for their valuable feedback.

---

### Meta-Review · Area_Chair_88rx · 2024-12-20

**Metareview:**

This paper offeres a practical, memory-efficient method for fine-tuning large language models on constrained hardware. All reviewers appreciate its novelty, clear presentation, and strong experimental results. The approach reduces memory usage during training while maintaining good performance at full scale. They highlight its potential to run 70B models on modest GPUs, which is very impressive. Though some initially questioned its generalizability and cost, the authors’ clarifications convinced most reviewers. Its a valuable improvement for researchers. Overall, the consensus is positive and the contribution is deemed worthy of acceptance.

**Additional Comments On Reviewer Discussion:**

Initial reviews raised concerns about: (1) limited novelty of combining pruning with LoRA, (2) fairness of comparisons and lack of baselines, (3) missing cost/throughput analysis, and (4) insufficient evidence for unchanged weights phenomenon. Authors addressed these by: providing detailed novelty justification, adding experiments with same-model comparisons, including comprehensive cost analysis showing comparable throughput to baselines, and adding evidence for unchanged weights in an appendix. Three reviewers raised their scores after authors' responses, acknowledging the contributions' significance. One reviewer maintained a low score, saying the method is a combination of existing methods and LoRA. The majority viewed the work as making valuable contributions to memory-efficient LLM training.

---

### Decision · Program_Chairs · 2025-01-22

Accept (Poster)